

# The effect of ash, water vapor, and heterogeneous chemistry on the evolution of a Pinatubo-size volcanic cloud

Mohamed Abdelkader[1], Georgiy Stenchikov[1], Andrea Pozzer[2], Holger Tost[3], and Jos Lelieveld[2]

[1]Division of Physical Sciences and Engineering, King Abdullah University of Science and Technology, Thuwal 23955-6900, Saudi Arabia
[2]Air Chemistry Department, Max Planck Institute for Chemistry, Mainz, 55128, Germany
[3]Institute for Atmospheric Physics, Johannes Gutenberg University of Mainz, Mainz, 55128, Germany

**Correspondence:** Mohamed Abdelkader (mohamed.ahmed@kaust.edu.sa)

**Abstract.** We employ the atmospheric chemistry general circulation model (EMAC) with gas phase, heterogeneous chemistry, and detailed aerosol microphysics to simulate the 1991 Pinatubo volcanic cloud. We explicitly account for the interaction of simultaneously injected $SO_2$, volcanic ash, and water vapor and conducted multiple ensemble simulations with different injection configurations to test the simulated $SO_2$, $SO_4^{2-}$, ash masses, stratospheric aerosol optical depth, surface area density
(SAD), and the stratospheric temperature response against available observations. We find that the $SO_2$, $SO_4^{2-}$ masses and stratospheric aerosol optical depth (SAOD) are sensitive to the initial height of the volcanic cloud. The volcanic cloud interacts with tropopause and stratopause, and its composition is shaped by heterogeneous chemistry coupled with the ozone cycle. The height of the volcanic cloud in our simulations is also affected by dynamic processes within the cloud, i.e., heating and lofting of volcanic products. The mass of the injected water vapor has a moderate effect on the cloud evolution when volcanic
materials are released in the lower stratosphere because it freezes and sediments as ice crystals. However, the injected water vapor at a higher altitude accelerates the oxidization of $SO_2$ which is sensitive to the injected water vapor mass (via hydroxyl production and reaction rate). The coarse ash comprises 98% of ash injection mass, which sediments within a few days, but aged sub-micron ash could stay in the stratosphere for a few months providing SAD for heterogeneous chemistry. The presence of ash accelerates the $SO_2$ oxidation that leads to a faster formation of the sulfate aerosol layer in the first two months after
the eruption and has to be accounted for in modeling the impact of large-scale volcanic injections on climate and stratospheric chemistry.

## 1   Introduction

Volcanic activity is a major natural cause of climate variation on both global and regional scales (Robock, 2000). Strong explosive volcanic eruptions inject a mixture of $SO_2$, volcanic ash, water vapor, halogens, and other tracers into the lower
stratosphere. The injected volcanic materials scatter and absorb incoming solar and outgoing terrestrial radiation, warming the stratosphere and cooling the Earth's surface and the lower troposphere (Hansen et al., 1992; Stenchikov et al., 1998; Kirchner et al., 1999; Robock, 2000; Soden, 2002; Shindell et al., 2001). Stratospheric warming (Stenchikov et al., 1998) and tropospheric cooling (Kirchner et al., 1999; Ramachandran et al., 2000) caused by the radiative impact of volcanic aerosols



yield to changes in atmospheric circulation, affect El Nino Southern Oscillation (ENSO) (Predybaylo et al., 2017), and force

a positive phase of the Arctic Oscillation (AO) (Graft et al., 1993; Kodera and Kuroda, 2000; Mao and Robock, 1998; Kodera and Kuroda, 2000; Stenchikov, 2002; Shindell, 2004; Stenchikov et al., 2006; Karpechko et al., 2010) causing boreal winter warming in middle and high latitudes over Eurasia and North America (Stenchikov et al., 2004; Thomas et al., 2009; Poberaj et al., 2011). The eruption of Mount Pinatubo (Philippines, 15 June 1991) with an Explosivity Index of VEI=6 caused the largest climate impact in the twentieth century. It is also by far the largest eruption to affect a densely populated area. The observed

global mean visible optical depth from the Pinatubo eruption reached 0.15. It was about two times higher than that of the second largest eruption in the $20^{th}$ century, El Chichon in 1982 (Dutton and Christy, 1992). The 1991 Mt. Pinatubo eruption is also the best observed explosive event with a detected significant climate impact. It has been documented by satellite instruments (McCormick, 1987; Long and Stowe, 1994), ground-based LIDARs and sunphotometers (Antuna et al., 2002, 2003; Good and Pyle, 2004; Nagai et al., 2010; Dutton and Christy, 1992; Thomason, 1992) , and airborne aerosol counters (McCormick et al.,

1995; Pueschel et al., 1994; Borrmann et al., 1995; Deshler, 2003). Mount Pinatubo produced about five cubic kilometers of dacitic magma. Three main volcanic explosions were reported to spread the volcanic ash and gases over 300,000 $km^2$.

The $SO_2$ mass emitted by the Mount Pinatubo eruption was estimated using Stratospheric Aerosol and Gas Experiment (SAGE), TOVS, TOMS and ground-based LIDAR retrievals (Guo et al., 2004a; Rose et al., 2006; Sheng et al., 2015; Krueger et al., 1995; Fisher et al., 2019). In addition to $SO_2$, Pinatubo injected tens of megatons of water vapor and volcanic ash into

the stratosphere (Guo et al., 2004a; Nedoluha et al., 1998; Joshi and Jones, 2009).

In the stratosphere $SO_2$ is oxidized by the OH radical to form sulfuric acid, which then binary nucleates in the presence of water to form sulfate aerosol. The primary source of OH in the stratosphere is ozone photolysis by ultraviolet radiation. This reaction forms oxygen and atomic oxygen in the excitation state ($O^{1d}$), which interacts with water vapor to form OH radicals. Thus, the $SO_2$ oxidation is controlled by the abundance of OH, which depends on the concentration of stratospheric water

vapor (Lovejoy et al., 1996). The co-injection of water vapor with $SO_2$ therefore accelerates the formation of sulfuric acid (LeGrande et al., 2016). The online calculation of OH is essential to correctly reproduce the dynamics of sulfate aerosol mass (Clyne et al., 2021; Stenchikov, 2021), and this has been neglected in many previous studies (Niemeier et al., 2009; Oman et al., 2006).

The sulfuric acid resulting from $SO_2$ oxidation nucleates to form long-lived sub-micron sulfate particles which interact with

solar and terrestrial radiation. The radiative effect and lifetime of sulfate aerosols depends on their size distribution, which is not definitively established. Therefore, different Pinatubo studies report a wide range of visible (0.5-0.6 μm) Stratospheric Aerosol Optical Depth (SAOD) for the same amount of injected $SO_2$. Brühl et al. (2015) obtained equatorial average SAOD=0.38 compared to SAOD=0.11 reported by LeGrande et al. (2016) for 17 Mt of injected $SO_2$. Niemeier et al. (2021) and Stenchikov et al. (2021) obtained similar SAOD which is consistent with observations for 17 Mt of injected $SO_2$. Dhomse et al. (2014),

using a detailed aerosol microphysics model, found that in simulations of a Pinatubo-like eruption with a 10 Mt of $SO_2$ injection, SAOD matches observations better than that with larger $SO_2$ emission. Mills et al. (2016) also reported that in their model a 10 Mt $SO_2$ injection produces the best fit to Pinatubo observations, while Sheng et al. (2015) and Sukhodolov et al. (2018) found that SAOD in their experiments with the emission of the 14 Mt of $SO_2$ best fits SAGE observations. Timmreck





et al. (2018) conducted ensembles of simulations with perturbed parameters, including the mass of injected SO2 and the
injection height of volcanic debris, to quantify the uncertainties in the radiative forcing of the 1991 Mt. Pinatubo eruption.

Volcanic ash (tephra) comprises silicate and volcanic glass with traces of gas bubbles (Kremser et al., 2016). Ash particles
have a wide range of sizes from sub-microns to millimeters (Rose and Durant, 2009) and highly irregular shapes. Large
ash particles with radii r>1 µm sediment relatively quickly (Niemeier et al., 2021, 2009; Stenchikov et al., 2021), and are
believed to contribute little in the long-term evolution of a volcanic cloud. Fine ash particles with r<1 µm disperse over
vast distances and can survive in the stratosphere for several months (Pueschel et al., 1994; Zhu et al., 2020; Russell et al.,
1996; Vernier et al., 2016), but their radiative effect is small because of their relatively smaller mass. However, Stenchikov
(2021) showed that despite the fact that most ash mass sediments during the first week after an eruption, ash solar and IR
heating and chemical/microphysical interactions with sulfate particles could affect the volcanic cloud formation and its long-
term evolution. Ash particles could be coated by sulfate, becoming chemically aged (Muser et al., 2020; Zhu et al., 2020).
They also uptake $SO_2$, thereby decreasing its abundance (Zhu et al., 2020). The coating and aging of ash particles increase
their size, alters their optical properties, and increase their deposition velocities (Muser et al., 2020; Zhu et al., 2020). The
enhanced ash sedimentation removes a portion of sulfate depending on the aging level. At the same time, stratospheric aerosol
particles (ash and sulfate) provide surfaces for heterogeneous chemical reactions affecting stratospheric chemical composition
(Muthers et al., 2015). Aerosol particles from volcanic eruptions increase the surface area density (SAD) and hence the rate
of heterogeneous reactions involving $ClONO_2$ and $N_2O_5$. This damps the $NO_x$ mixing ratios altering the OH stratospheric
budget, which affects the rate of $SO_2$ oxidation (Prather, 1992; Kilian et al., 2020). Fig. 1 summarizes the microphysical and
chemical transformations of the erupted volcanic materials.

Like mineral dust, volcanic ash absorbs and scatters solar shortwave (SW) and terrestrial longwave (LW) radiation. This has
a significant impact on the chemistry and radiation budget of the atmosphere in the first few days after an eruption, causing
rapid lifting of volcanic debris (Stenchikov et al., 2021; Niemeier et al., 2009). The models with different physics calculate the
evolution of volcanic clouds and their impact on climate assume various $SO_2$ injection heights, initial plume composition (ash
and water are often not injected), different spatial-temporal resolutions, different treatment of ash-sulfate interaction, and ash
chemical aging. The differences in physics and chemistry translate into the differences in volcanic cloud evolution and radiative
effect. For instance, those models which generate finer sulfate particles (Mills et al., 2017; Dhomse et al., 2020) overestimate
the stratospheric sulfate lifetime, generating higher SAOD for the same $SO_2$ injection mass.

Along with the $SO_2$ mass, the injection height of volcanic debris is a critical parameter for correctly simulating the dispersion
of a volcanic cloud, as it associates with the wind field that transports the volcanic plume. Sheng et al. (2015) performed a
sensitivity study for the initial mass and altitude of the injected $SO_2$ for the Pinatubo eruption and showed that a mass of 17 Mt
of $SO_2$ or less gives the best agreement with the SAGE optical depth within a peak of the volcanic cloud between 18-21 km.
The transient equilibrium height of the volcanic plume depends not only on the height of initial injection but also on internal
feedback mechanisms. Stenchikov et al. (2021) demonstrated that radiative heating by ash was lifting volcanic debris by 1 km
per day during the first week following the 1991 Pinatubo eruption. Muser et al. (2020) reported the lifting of a volcanic plume
of Raikoke eruption that was an order of magnitude smaller than the 1991 Pinatubo eruption. Volcanic debris injections cause





significantly different localized radiative heating and lofting when quasi-zonal, as in Brühl et al. (2015) and when localized, as in LeGrande et al. (2016). In contrast to Stenchikov et al. (2021), we explicitly calculate ash chemical aging, stratospheric ozone chemistry, and aerosol microphysical processes.

The underlying dynamic and/or chemical mechanism of the large sensitivity of SAOD to the injection height has not been recognized yet in a fully interactive model. The effects of injected "volcanic" water and chemical aging of volcanic ash on $SO_2$ oxidation rate and $SO_4^{2-}$ removal are not studied within the models with comprehensive gaseous and heterogeneous chemistry

and detailed microphysics. Here we use the Atmospheric Chemistry and Aerosol General Circulation Model (EMAC) with multi-phase chemistry along with detailed aerosol microphysics to study the evolution of a Pinatubo-size volcanic cloud. We account for the entire range of dynamic, chemical, and microphysical complexity of the process to address the following science questions:

– How does the initial spatial distribution and height of injected volcanic debris affect the evolution of a volcanic cloud?

– What is the effect of heterogeneous chemistry on the $SO_2$ oxidation rate within a volcanic cloud?

– How do co-injection of $SO_2$, water vapor, and ash, affect volcanic cloud evolution?

– How does the aging of co-injected ash affect volcanic cloud development?

## 2 Data

To constrain the simulations and evaluate the model results, we use the SAGE data set with partially filled gaps compiled by

Stratospheric Processes and its Role in Climate (SPARC) and published in the Assessment of Stratospheric Aerosol Properties (ASAP) report (Thomason and Peter, 2006). This data set provides the aerosol effective radius and aerosol extinction in UV, visible (0.525 μm), and near IR (1.02 μm) wavelengths. The SAGE/ASAP SAOD is zonal mean and collected on a monthly basis. It is available from 70°S to 70°N with 5° resolution in latitude between 1984 and 1999. We further refer to this data as SAGE/ASAP SAOD or $R_{eff}$. The SAGE observations of aerosol extinction contain multiple gaps at the initial stage of the

volcanic cloud evolution because of the instrument's saturation (Thomason, 1992). The observations in near IR are of better quality than in visible or UV (Stenchikov et al., 1998). Therefore, to obtain the visible SAGE/ASAP SAOD, we scale near IR SAOD using the angstrom exponent obtained from our simulations, similar to (Stenchikov et al., 1998). We also use the Advanced Very High-Resolution Radiometer (AVHRR) SAOD at 0.63 μm (Long and Stowe, 1994). The AVHRR observations are collected over the oceans at 0.1°x0.1° horizontal resolution for cloud-free conditions at daytime. The AVHRR AOD is

measured for the entire atmospheric column including the troposphere. To obtain the AVHRR stratospheric AOD, we calculate the AVHRR AOD monthly climatology for the pre-Pinatubo period of 1985-1990 and subtract it from the total AOD for the Pinatubo period. Unfortunately, this can introduce some level of uncertainty due to the high variability of tropospheric AOD. We refer to the visible SAOD obtained from SAGE as the scaled SAGE/ASAP SAOD. We compare scaled SAGE/ASAP and AVHRR SAOD at 0.63 μm to the model visible SAOD.



Krueger et al. (1995) estimated the mass of $SO_2$ during the first 15 days after the 1991 Pinatubo eruption based on the Total Ozone Mapping Spectrometer (TOMS) observations. They concluded that the mass of initially emitted $SO_2$ was 15±3 Mt. Guo et al. (2004a) later estimated the emitted mass of $SO_2$ to be 14-20 Mt. Recent estimates reduce the initial $SO_2$ mass to 12 Mt (Fisher et al., 2019). Estimates of $SO_2$ mass using retrievals from the Optical Vertical Sounder/High-Resolution Infrared Radiation Sounder/2 (TOVS) on the Television Infrared Observation Satellite (TIROS) suggest that the initial $SO_2$ mass was

19±4 Mt (Guo et al., 2004a). However, the TOVS retrievals are less accurate than TOMS because they are affected by sulfate aerosol absorption in IR. $SO_4^{2-}$ mass has also been estimated using the High-Resolution Infrared Radiation Sounder (HIRS) (Guo et al., 2004a). However, the estimated sulfate aerosol mass depends on the aerosol size distribution, which is not well known, and this introduces uncertainties into estimated $SO_4^{2-}$ mass.

Volcanic ash mass is available for the first few days after the Pinatubo eruption from AVHRR and High-Resolution Infrared

Radiation Sounder/2 (HIRS/2) observations (Guo et al., 2004b). HIRS/2 detected 80Mt of fine ash in the atmosphere on the first day after the eruption. AVHRR ash retrievals evaluate the spectral contrast of radiance (Aerosol Index) to distinguish between absorbing aerosols, such as volcanic ash, and non-absorbing aerosols, such as sulfate. However, the retrieval algorithm does not consider particles smaller than 1 μm (Guo et al., 2004b).

We obtain the stratospheric temperature response to the 1991 Pinatubo eruption from the MERRA2 reanalysis data available

on 0.5°x0.625° horizontal grid and 72 vertical levels from the surface to 0.01 hPa (Gelaro et al., 2017). To reproduce the effect of the 1991 Pinatubo eruption, MERRA2 assimilates observations from different satellite sensors such as TOVS and the Spinning Enhanced Visible and InfraRed Imager (SEVIRI) on the Meteosat Second Generation (MSG) Satellite, as well as the Microwave Limb Sounder (MLS). The MERRA2 temperature anomalies were calculated with respect to 1985-1990 climatology.

## 145 3 Model

Here we employ the ECHAM5/MESSy2 atmospheric chemistry model, EMAC (Joeckel et al., 2005, 2006, 2010). EMAC is a modular model based on sub-models that describe processes in the stratosphere, the middle atmosphere, and the troposphere, accounting for anthropogenic emissions and interactions with oceans and land (Joeckel et al., 2010). EMAC has been used to study impacts of volcanic stratospheric aerosols on climate and stratospheric circulation (Brühl et al., 2012, 2015; Bingen

et al., 2017; Löffler et al., 2016; Kilian et al., 2020, among others), as well as dust aging and dust-air pollution interaction in the troposphere (Abdelkader et al., 2015, 2017; Klingmüller et al., 2019, 2020).

The Modular Earth Submodel System (MESSy) links the various submodels. The submodels comprise AEROPT, CLOUD, CONVECT, CVTRANS, DDEP, GMXE, JVAL, LNOX, MECCA, OFFEMIS, ONEMIS, RAD4ALL, SCAV, SEDI, TNUDGE, and TROPOP. Table 1 shows the submodels used in this study indicating their functionality, while the detailed description of

all EMAC submodels can be found in Joeckel et al. (2010). We configure EMAC using MESSy version 2.52 with the $5^{th}$ generation European Centre Hamburg Atmospheric general circulation Model version 5.3.02, ECHAM5 (Roeckner et al., 2006), and employ the same chemistry and aerosol microphysics setup as in (Brühl et al., 2012, 2015). For vertical approximation



we employ 90 sigma-hybrid levels from the earth surface up to 0.01 hPa, and T42 spectral approximation horizontally that corresponds to 2.8° grid spacing at the equator both in longitude and latitude. Varying monthly sea surface temperature and
sea ice are prescribed from AMIPII dataset (Taylor et al., 2000).

We apply the quasi-biennial oscillation (QBO) submodel to capture the observed phase of QBO and account for its effect on the stratospheric circulation, similar to Stenchikov et al. (2004). No other constraints are imposed on the model dynamics, e.g., we do not nudge tropospheric winds.

The emission inventory comprises the sources of greenhouse gases, $NO_x$, CO, NMVOCs, $NH_3$, $SO_2$, black carbon (BC),
and organic carbon (OC). The emissions are monthly mean and geographically distributed according to the EDGAR4 2009 emission inventory and the Global Fire Emissions Database (GFED) version 3 (van der Werf et al., 2010). We also account for the DMS and OCS emissions similar to Brühl et al. (2015). To calculate atmospheric composition, we employ 230 gas-phase chemical reactions, 76 photolytic reactions, and 12 heterogeneous reactions for 159 species. The photolysis rates are calculated within the model for the spectral range $178.6\,nm \leq \lambda \leq 752.5\,nm$ accounting for gaseous absorption ($O_3$ and $O_2$), Rayleigh
scattering, absorption, and scattering by aerosols and clouds (Landgraf and Crutzen, 1998; Sander et al., 2011). In this setup, the photolysis rates are not coupled to volcanic aerosol. The model calculates the instantaneous radiative forcing using double radiation calls, with and without aerosols. Aerosol microphysics and chemistry are called every model time step, while the radiation sub-model is called every third-time step.

### 3.1 Stratospheric sulfate chemistry

Volcanic sulfate results from the oxidation of $SO_2$ by OH in the presence of water vapor. OH is produced by ozone photolysis by UV radiation with wavelengths less than 0.242 μm. This reaction forms $O_2$ and excited oxygen $O(^{1d})$ (Eq. R1). The excited oxygen radical interacts with water to form the hydroxyl radical OH (Eq. R2), which oxidizes $SO_2$ in two steps to form sulfate. At the first step, OH oxidizes $SO_2$ to form $SO_3$ and $HO_2$ (Eq. R3). At the second step, $SO_3$ interacts with water molecules to form sulphuric acid (Eq. R4). The rate of reaction in Eq. R4 depends on the concentration of water molecules that are also
in the reactants (Burkholder et al., 2015). Therefore, higher water vapor concentrations significantly increase the formation rate of sulphuric acid. New sulfate particles are generated by the binary nucleation of sulfuric acid and water molecules. Thus, the formation of sulfate particles in a volcanic cloud depends strongly on water vapor concentration. The models that do not parameterize nucleation explicitly are less sensitive to the abundance of water vapor in a volcanic cloud than those that do (LeGrande et al., 2016).

$$O_3 \xrightarrow[<310nm]{h\nu} O(^1d) + O_2 \tag{R1}$$

$$O(^1d) + H_2O \rightarrow 2OH \tag{R2}$$


$$SO_2 + OH \rightarrow SO_3 + HO_2 \tag{R3}$$

$$SO_3 + H_2O \xrightarrow[8.5E^{-41} \exp^{\frac{6540}{T}}]{C[H_2O]} H_2SO_4 \tag{R4}$$

$$H_2SO_4 \xrightarrow{\text{nucleation}} SO_4^{2-} + 2H^+ \tag{R5}$$

### 3.2 Aerosol Microphysics

The aerosol setup in EMAC has been described in detail in (Pringle et al., 2010; Tost et al., 2010; de Meij et al., 2012; Pozzer et al., 2012; Brühl et al., 2015; Abdelkader et al., 2015, 2017). We use the aerosol microphysics sub-model GMXe (Pringle et al., 2010), coupled to the gas-aerosol partitioning scheme ISORROPIA-II (Fountoukis and Nenes, 2007) and heterogeneous chemistry scheme (Sander et al., 2005, 2011). Aerosol size distributions in the model are approximated by seven lognormal modes: four soluble modes (nucleation, Aitken, accumulation, coarse) and three insoluble modes (Aitken, accumulation, coarse). In our simulations, sulfate represents by the soluble modes, and ash is initially considered insoluble until it ages, i.e., five monolayers of sulfate particles coat the ash particle. The modes' median radii change in time during aerosol microphysical transformations, but the widths of the modes remain fixed. The median radii for three insoluble modes and dry cores of four soluble modes initially are equal to 0.0015, 0.025, 0.25, and 2.5 µm for nucleation, Aitken, accumulation, and coarse modes, respectively. The widths of the lognormal distributions for the above modes are 1.59, 1.59, 1.49, and 1.70, respectively (Brühl et al., 2015).

Aerosols in soluble modes evolve by uptake or loss of water and $SO_4^{2-}$ molecules, and coagulation. The hygroscopic growth of ash is only allowed in a soluble mode (Abdelkader et al., 2015). The mass of large or fine aerosol particles in the distribution tails is assigned to a corresponding neighboring mode when the mode's median radius reaches a certain threshold. The aerosol modes and the thresholds are schematically shown in Fig. 2. In our simulations, we choose threshold radii equal to 0.0005, 0.006, 0.07, 1.6 µm for the nucleation, Aiken, accumulation, and coarse modes respectively, as in (Brühl et al., 2015).

### 3.3 Volcanic ash - Model Implementation

We introduce a new "ash" tracer to account for volcanic ash. We assume the ash density to be 2400 kg m$^{-3}$, similar to that of mineral dust, as ash comprises mainly silicate ($SiO_2$). Therefore, for calculating chemical aging we assume that ash particles have the same water uptake and accommodation coefficients as dust particles (Abdelkader et al., 2015).

High-density ash particles sediment faster than pumice assumed in (Zhu et al., 2020). Zhu et al. (2020) considered the Kelud eruption that emitted 100 times less volcanic material than the 1991 Pinatubo eruption. Stenchikov et al. (2021) showed that applying the assumption about long-lived ash for the larger volcanic explosions like the 1991 Pinatubo eruption could cause unrealistic overheating of the stratosphere.



For a full representation of chemical aging, we use a comprehensive chemistry scheme that enables the production of the primary inorganic acids which contribute to the chemical aging of ash particles (Metzger et al., 2016).

Volcanic ash is removed from the stratosphere mainly by gravitational sedimentation. Sedimentation parameterization in EMAC utilizes the Walcek scheme (Walcek, 2000; Kerkweg et al., 2006a). Ash scavenging is implemented in EMAC by Tost et al. (2006a) and is fully coupled with the aerosol and gas-phase chemistry.

To calculate the optical properties of volcanic ash, we choose its complex refractive index to be equal to that of dust assuming ash particles absorb solar and terrestrial radiation (Pollack et al., 1973; Vogel et al., 2017; Stenchikov et al., 2021). In visible light the ash refractive index RI=1.53 + 0.004$i$. Ash is more absorbing in UV, near-infrared (NIR), and Infrared (IR) than in visible. Table 1 in the supplement shows the volcanic ash refractive index as a function of wavelength. Fig. 8 shows the refractive indices used in EMAC model for different aerosols as a function of wavelength.

### 3.4   Aerosol Radiative Effect

We use the AEROPT submodel to calculate extinction, single-scattering albedo, and asymmetry parameter, the aerosol optical properties required for the radiative transfer calculations. It is assumed that different types of aerosols are mixed internally so that the refractive index of the mixture is calculated from the volume fractions of the aerosol components. The sensitivity to this assumption is discussed in detail by Klingmüller et al. (2014). The optical properties are calculated for each aerosol mode

independently. The RAD submodel calculates radiative transfer (Roeckner and Coauthors, 2003). The Fouquart and Bonnel scheme (Fouquart and Bonnel, 1980) is used for calculating shortwave radiation, while longwave radiation is calculated using RRTM (Iacono et al., 2008). Scattering of the infrared radiation by aerosols is neglected. RAD accounts for shortwave and longwave absorption of water vapor, clouds, $O_3$, $CH_4$, N2O, CO2, CFCs, and aerosols, including sulfate and volcanic ash implemented in this study. Table 2 shows the shortwave and longwave bands used in the radiative transfer calculations in

EMAC. For comparison with observations, we consider the first two SW bands in Table 2 as visible and near-infrared.

### 4   Experimental Setup

The complete set of experiments is listed in Table 3. The control experiment (ctrl) describes the state of the atmosphere from 1990 to 2000 when unperturbed by volcanic eruption. All perturbed simulations (those with volcanic aerosols present) were conducted from June 1, 1991, to December 31, 1994. We emit 17 Mt of sulfur dioxide in all perturbed simulations, with the

exception of one experiment specifically marked and used to study sensitivity to $SO_2$ emission mass.

Along with $SO_2$, we consider co-injections of water vapor and ash. For ash, we adopt the same initial size distribution as in (Niemeier et al., 2009) and (Stenchikov et al., 2021). We redistribute the total emitted fine ash mass of 75 Mt (Guo et al., 2004b) between two insoluble modes, accumulation and coarse (Fig. 2). The accumulation mode comprises 1.5 Mt of ash, and the coarse mode comprises 73.5 Mt of ash. The ash mass in the accumulation mode is important since it has a much longer

lifetime than ash in the coarse mode. We use the standard EMAC *import* and *offemis* submodels to initialize the $SO_2$, water





vapor, and ash tracers (Kerkweg et al., 2006b). Section 1 in the supplement explains the implementation of $SO_2$, water vapor and volcanic ash injection mechanism in EMAC.

In the main set of experiments, we release volcanic products in the specified model grid box centered at the 17 km, 20 km, or 25 km height at the geographical coordinates of Mt. Pinatubo (15.1429 °N, 120.3496 °E) with pre-calculated emission
rates (in molecules $m^{-3}\,s^{-1}$) during 24 h. Here we refer to these as a one-grid-box emission scheme. See Table 3 for details. In the 1s1-17km, 1s1-20km, and 1s1-25km experiments, we assume that only $SO_2$ is injected at 17 km, 20 km or 25 km, respectively. In the 1w1-20km experiment we release $SO_2$ and water vapor (Nedoluha et al., 1998; Joshi and Jones, 2009) at 20 km. The va0 experiments employ the same settings as 1w1 but assume injection of 75 Mt of ash. The va0 experiments do not account for the chemical aging of ash. The va1 experiments are similar to va0 but account for ash aging. In contrast
to the one-grid-box emission scheme, in experiment 3s10-25km we inject $SO_2$ in the 3000 km wide latitude belt centered at the latitude of the eruption mimicking the setting in Brühl et al. (2015). The injected layer is 10 boxes thick (5 km) and is centered at the altitude of 25 km. The experiments are listed in Table 3. We refer to the clusters of experiments with the same physics using a generic name without specifying injection altitude, such as 1s1, 1w1, va0, va1, in instances when this will not cause confusion. Experiments 1s1 are used to study the sensitivity to the height of the injection of volcanic $SO_2$. The 1w1
experiments with 150 Mt and 15 Mt injected water allow us to quantify the dependence of the mass of injected water vapor. Experiments va0 and va1 are designed to quantify the effect of ash and ash aging, respectively. Experiment 3s10-25km mimics the quasi-zonal $SO_2$ injection from Brühl et al. (2015). Experiment va1-20km-12Mt is designed to study dependence on the amount of injected $SO_2$.

All simulations are conducted with a one-year spin-up not included in the analysis. To reduce the effect of internal model
variability in each experiment, we calculate five ensemble members using different atmospheric initial conditions. The analysis in this study is performed and presented for the ensemble means.

## 5 Results

First, we compare the model results with observations, focusing on spatial-temporal distributions of $SO_2$, $SO_4^{2-}$ and other related chemicals. We also compare the stratospheric AOD (SAOD) which defines volcanic radiative effect, and the stratospheric
temperature response which measures volcanic climate impact. In addition, we compare the Surface Area Density (SAD) that controls heterogeneous chemistry within the volcanic cloud, and aerosol effective radius ($R_{eff}$) that characterizes aerosol size distribution (see Fig. 3-6). The spatially averaged $R_{eff}$ is calculated as a ratio of the third $M3_m$ and the second $M2_m$ moments of each aerosol mode $m$ integrated over the entire domain (Eq. 1 and Eq. 2). The effective radii for individual modes and for the entire aerosol size distribution are given by (Eq. 3) and (Eq. 4), respectively.

$$M2_m = \iiint_v N_m R_m^2 exp^{(2ln^2\sigma_m)} dx dy dz \qquad (1)$$



$$M3_m = \iiint\limits_{v} N_m R_m^3 exp^{(\frac{9}{2}ln^2\sigma_m)} dxdydz \tag{2}$$

$$R_{eff}^m = \frac{M3_m}{M2_m} \tag{3}$$

$$R_{eff}^{total} = \frac{\sum_{m=1}^{m=Nmodes} M3_m}{\sum_{m=1}^{m=Nmodes} M2_m} \tag{4}$$

Where $N_m$ is the number density for aerosol mode $m$, $R_m$ is the median radius, and $\sigma_m$ is the width of the aerosol mode $m$. $N_{modes}$ is the number of aerosol modes.

Figures 3-5 show various parameters in 3s10-25km, 1s1 experiments with the different injection heights, as well as AVHRR, and SAGE/ASAP observations. The AVHRR zonal mean visible SAOD is largely consistent in spatial-temporal behavior with the scaled SAGE/ASAP SAOD (Fig. 4). The original SAGE/ASAP visible SAOD is almost half of the AVHRR SAOD, because the SAGE II sensor was saturated during the few first weeks after the eruption, therefore data at the initial stage of eruption are sparse. The AVHRR continuously sensed the entire atmospheric column including troposphere, the effect of which could be estimated only approximately (Thomason, 1992; Russell et al., 1996; Kremser et al., 2016). The consistency between the scaled SAGE/ASAP and AVHRR visible SAODs lessens in late fall of 1991, when scaled SAGE/ASAP SAOD begins overestimating AVHRR SAOD. Discrepancies between different data sets are discussed in (Bingen et al., 2004). Despite sparse observations at the initial stage of volcanic cloud development, SAGE/ASAP is the only global satellite observation that recorded the vertical structure of the Pinatubo cloud. For example, Fig. 5 demonstrates aerosol SAD at different altitudes as reported by SAGE/ASAP and simulated in the model.

Below we study the sensitivity of volcanic cloud evolution to all the main factors: injection height, amount of injected water, injection of ash, and ash aging. We start from sensitivity to injection height using the simplest 1s1 experiments with $SO_2$ only injections. The cloud height is essential because it defines the wind field that drives cloud dispersion. The $O_3$ mixing ratio and abundance of water vapor, which affect chemical and microphysics transformations within the plume are also height dependent.

## 5.1 Sensitivity to Injection Height

Figure 3a,b,c compares the observed and simulated SAOD, $SO_2$ and $SO_4^{2-}$ masses, and $R_{eff}$ in the 1s1 experiments with different injection heights, respectively. The altitudes where volcanic debris resides depend not only on the initial injection height but upward stratospheric motion and lofting driven by radiative heating of volcanic debris (Stenchikov, 2021; Niemeier et al., 2009; Kinnison et al., 1994; Aquila et al., 2012). The latter process and the rate of chemical transformations within a volcanic cloud are sensitive to initial concentrations of optically and chemically active materials within a fresh volcanic cloud, i.e., in terms of our simulation settings, from the volume that a cloud initially occupies.



Experiment 3s10-25km assumes a zonally uniform $SO_2$ release at 25 km altitude within a latitude belt centered at the latitude of the eruption (15.1429°N). The visible SAOD from this experiment compares well with observations. In experiment 1s1-25km, we release $SO_2$ centered at the same height, as in the 3s10-25km experiment, but within one model grid box at the geographic coordinates of the Pinatubo eruption (15.1429°N, 120.3496°E). This causes initially higher $SO_2$ concentrations compared to the 3s10-25km experiment. The volcanic debris is released with a constant mass emission rate and spread for

more than 1000 km during the 24 hours of emission. Despite that $SO_2$ was released at the same altitude, these two experiments exhibit remarkable differences in the SAODs (see Fig. 3a), $SO_4^{2-}$ masses (Fig. 3b), and spatial distributions of SAOD and SAD (Fig. 4 and Fig. 5). To understand the mechanism of the strong sensitivity of the volcanic cloud evolution to its initial stage, below we test the 3s10-25km experiment and the one-grid-cell $SO_2$-only injection experiments 1s1 with 17 km, 20 km, and 25 km injection heights against observations.

### 5.1.1 SAOD

Contrary to the 1s1-25km experiment, the visible tropical SAOD in experiment 1s1-20km compares well with that from the scaled SAGE/ASAP and AVHRR observations (Fig. 3a). The visible SAOD from the Sixth Coupled Model Intercomparison Project (CMIP6) (Eyring et al., 2016; Zanchettin et al., 2016) which mimics the original visible SAGE/ASAP extinctions develops slowly and is half of scaled SAGE/ASAP and AVHRR.

The equatorial average (20S-20N) SAOD in 1s1-17km is half the size of 1s1-20km and 3s10-25km. The 1s1-25km SAOD is even smaller (Fig. 3a). All SAODs except that in the 1s1-25km experiment are bigger than the CMIP6 SAOD. The SAOD in the 3s10-25m experiment grows faster than in 1s1 experiments, reaching 0.33 in August 1991. At a given chemical composition of sulfate aerosol particles, the transient SAOD depends both on the $SO_4^{2-}$ mass, i.e., the rate of oxidation of $SO_2$ to $SO_4^{2-}$, and on aerosol size distribution, i.e., $R_{eff}$. The smaller sulfate aerosol particles have a bigger collective cross-section per unit mass

than larger ones. So a bigger mass of large sulfate particles might have a smaller SAOD than a smaller mass of smaller sulfate particles. This must be considered when evaluating the mass of $SO_4^{2-}$ and the sulfate aerosol SAOD in observations and model experiments.

### 5.1.2 Oxidation of $SO_2$

Figure 3b shows the globally integrated $SO_2$ and $SO_4^{2-}$ masses in the 3s10 and 1s1 experiments with the different emission

heights as functions of time. The $SO_2$ mass in the 1s1-20km experiment decreases more slowly than in all other experiments. Furthermore, the $SO_4^{2-}$ mass in the 3s10-25km grows faster than in the other experiments in Fig. 3b. This is because the $SO_2$ oxidation rate depends on the abundance of OH radicals. The OH production depends on $O_3$ concentration and incoming UV radiation. Because $SO_2$ is distributed zonally over the entire latitude belt in the 3s10-25km experiment, its concentration in a volcanic cloud is lower than in all one-grid-box-injection experiments. Hence, the $SO_2$ oxidation is more efficient in the

3s10-25km experiment because there are more OH radicals available per $SO_2$ molecule in the latitude belt than in a smaller volcanic cloud volume as in the one-grid-box experiments. Furthermore, OH is less depleted by $SO_2$ in a larger volume. All 1s1 experiments underestimate $SO_4^{2-}$ mass in the first few days in comparison with the available observations (Fig. 3b). The





presence of $SO_4^{2-}$ in a fresh volcanic plume detected in observations is confusing as the models usually do not account for the physical mechanisms that could produce it in such a short time. To explain this discrepancy, Guo et al. (2004a) suggested that

1-2 Mt of $SO_4^{2-}$ was injected at the initial stage of the eruption. However, we do not account for the initial $SO_4^{2-}$ release in this study.

### 5.1.3   Spatial-temporal Evolution of SAOD and SAD

The spatial-temporal patterns of visible SAOD in the 1s1-20km experiment compare well with AVHRR and scaled SAGE/ASAP observations (Fig. 4), although the aerosol poleward transport in the model is too fast. This is a known deficiency of global

models which simulate subtropical barriers which are too transparent due to coarse spatial resolution. The 1s1-25km visible SAOD is smaller than the scaled SAGE/ASAP and AVHRR SAODs, and exhibits qualitatively incorrect evolution of the volcanic cloud which moves too far north, similar to that reported by Stenchikov et al. (2021) for volcanic injection at 24 km altitude. The 3s10-25km SAOD has a realistic spatial-temporal structure but substantially overestimates observed SAODs. SAOD in the 1s1-17km experiment (Fig. 4) exhibits even faster poleward transport than in the 1s1-20km run due to stronger

wave activity at lower altitudes in the stratosphere. In this experiment, the equatorial aerosol reservoir dissipates too quickly because of its proximity to the tropopause and too intensive poleward transport.

Figure 5 compares the SAD in the 1s1-17km, 1s1-20km, and 1s1-25km experiments with the SAGE/ASAP observations (Thomason et al., 1997). SAD facilitates heterogeneous reactions in the volcanic cloud. Both sulfate aerosols and volcanic ash contribute to SAD, but in 1s1 experiments, we only account for sulfate aerosol surfaces. Therefore it is expected that the

simulated SAD will be smaller than the observed one, especially at the very beginning after the eruption. Only the 1s1-20km experiment shows SAD distributions consistent with observations at all three levels; 20 km, 25 km, and 30 km. Both the model and SAGE/ASAP show that the peak SAD is at 20 km. At higher altitudes (25 km and 30 km), SAD is smaller than at 20 km altitude. This suggests that volcanic material in the simulations has been lifted by at least 5 km above the injection level. In the 1s1-17km experiment the model underestimates SAD at 25 km and 30 km, while the volcanic cloud remains at and below

20 km level. In the 1s1-25km experiment, the volcanic cloud resides unrealistically high, at and above 30 km. At that height sulfate droplets tend to evaporate and the sulfuric acid photolyzes back to $SO_2$, and is eventually transported to the mesosphere (Rinsland et al., 1995).

### 5.1.4   Aerosol Size Distribution

Figure 3c compares $R_{eff}$ from SAGE/ASAP averaged over the tropical belt, the 3S10-25km, and the 1s1 experiments with

17, 20, and 25 km injection heights. In the control case, the model $R_{eff}$=0.14 μm is lower than the observed unperturbed value of 0.17-0.19 μm (Russell et al., 1993), as the model underestimates the effect of anthropogenic sulfur emissions on the stratospheric Junge layer (Marandino et al., 2013; Brühl et al., 2015).

SAGE-II observations suggest that aerosol extinction increases, and its maximum shifts from 0.385 μm to 525 μm soon after the Pinatubo eruption, indicating the sudden increase of sizes of aerosol particles (Thomason, 1992; Thomason and Peter,



2006; Kremser et al., 2016). The observed effective radius increases from the background level to about 0.5 μm in six months
        (Russell et al., 1996).

        In Figure 3c, $R_{eff}$ in the 1s1-20km experiment increases gradually, reaching maximum $R_{eff}$ = 0.4 μm in September of
        1991, and then decreases due to settling of larger particles. In the 1s1-25km experiment, $R_{eff}$ is the largest, when compared
        with other experiments, as it generates the largest $H_2SO_4$ concentration. It initially grows faster than in all other runs, reaching
maximum $R_{eff}$ = 0.45 μm in August of 1991, and then decreases, merging with all other experiments in December of 1991. In
        experiment 1s1-17km $R_{eff}$ is the smallest when compared to other experiments, as it looses $SO_2$ mass through the tropopause.
        All the simulations are predicting quite similar temporal evolution of $R_{eff}$.

        The tropical visible SAODs in Fig. 3a are consistent with $SO_4^{2-}$ mass and $R_{eff}$. That is, the $SO_4^{2-}$ mass in the 1s1-25km
        experiment is larger than in the 1s1-17km experiment, but SAOD is smaller because 1s1-25km $R_{eff}$ is bigger. The simulated
$R_{eff}$ in 1s1-25km and 3s10-25km have maximums above that in SAGE/ASAP. The $R_{eff}$ in the 1s1-17km and 1s1-20km runs
        is always below the SAGE/ASAP $R_{eff}$. However the SAGE/ASAP $R_{eff}$ is itself quite uncertain (Ansmann et al., 1997).

### 5.1.5    Impact on Chemical Composition

        Figure 6 shows vertical cross-sections of the mixing ratios or concentrations of $SO_2$, $SO_4^{2-}$, OH, $H_2SO_4$, $NO_x$, $NO_y$ in
        the 1s1 experiments with 17km, 20km, and 25km injection heights (see Eq. R1 - Eq. R5) averaged over the equatorial belt
(20°S - 20°N). We do not account here for the $SO_2$ radiative effect (Stenchikov, 2021), and there is no ash injection in these
        experiments. Therefore it is only sulfate aerosols that cause heating and lofting of the volcanic cloud. After the injection,
        lifting is caused by regional upward motion in the Brewer–Dobson circulation before $SO_4^{2-}$ develops, being reproduced by
        EMAC. The $SO_2$ and $SO_4^{2-}$ clouds separate due to gravitational settling of sulfate aerosols (Fig. 6a-f). This initiates multi-layer
        distributions of all other tracers. Stratospheric vertical uplift depends on the altitude which is getting stronger at higher altitude
(at least in EMAC). This is well seen in the 1s1-25km run in comparison with the 1s1-17km and 1s1-20km runs (see Fig. 6a-c).

        The volcanic cloud in the 1s1-25km experiment rises to 30 km, significantly higher than all other experiments. This affects
        the development of the volcanic cloud. The $SO_2$ oxidation rate slows down as the temperature rises. The $SO_4^{2-}$ mass is therefore
        smaller than in the other experiments. In addition, in the 1s1-25km experiment $R_{eff}$ is higher than in other experiments. This
        factor tends to lower the SAOD, since larger particles in the 1s1-25km experiment are less optically efficient per unit mass, and
have a lower lifetime with respect to gravitational settling. Therefore, SAOD and SAD in this experiment are smaller than in
        the others in Fig. 3 and Fig. 5.

        The experiments with different emission heights result in differences in the SAOD in Fig. 3 and SAD in Fig. 5. This partially
        results from different $SO_2$ oxidation rates that are defined by the abundance of OH radicals at different altitudes. Oxidation of
        volcanic $SO_2$ in the stratosphere also perturbs the Chapman cycle and reduces the ozone mixing ratio in the stratosphere.
Three weeks after the eruption, OH is reduced around the injection height because of stratospheric water consumption by
        the oxidation of $SO_2$ in all three experiments in Fig. 6g,h,i. OH remains depleted above the $SO_4^{2-}$ cloud, where $SO_2$ mixing
        ratio is high. The change in OH is generally largest in the 1s1-25km experiment.





The increase of $H_2SO_4$ is also more pronounced in the 1s1-25km experiment (Fig. 6j,k,l). Initially, the $H_2SO_4$ increase develops at the emission level. This is seen until November of 1991. Then a secondary plume of $H_2SO_4$ is formed at a higher
altitude, above 29km.

We account for twelve heterogeneous reactions. Following Danilin et al. (1999), we evaluate the effect of the heterogeneous reactions by the abundance of $NO_x$ ($NO+NO_2$) and total inorganic nitrogen, $NO_y$ ($NO_x + NO_3 + HNO_3 + 2*N_2O_5 + HONO + HNO_4 + ClONO_2 + BrONO_2$).

The heterogeneous chemistry might also influence the oxidation capacity by chlorine and bromine activation; however, as
no additional halogen emissions from the eruption are considered, this effect might be minor.

Figure 6m-r shows the strong dependence of $NO_x$ and $NO_y$ on injection height within the aerosol cloud. The $NO_x$ mixing ratio decreases, and the $NO_y$ mixing ratio increases, along with the increase of the injection height. The changes in $NO_x$ and $NO_y$ affect the ozone cycle (Seinfeld and Pandis, 2006). The dependence of the background ozone concatenations on altitude adds to the sensitivity of the cloud evolution to injection height. Furthermore, the modified ozone concentrations feed back to
the OH production and hence the sulfur oxidation.

Along with the chemical processes, the interaction of volcanic debris with the tropopause and the stratopause, adds in sensitivity of the $SO_4^{2-}$ mass to the height of the injection. In the $17\,\mathrm{km}$ injection height experiments, the cloud loses part of the mass through the tropopause, but in the $25\,\mathrm{km}$ injection height experiment, part of the sulfur is transported to the mesosphere and lost for immediate sulfate formation. It descends to the stratosphere again in high latitudes winter. The volcanic debris
injected at $20\,\mathrm{km}$ stabilizes in the middle of the stratosphere. Hence it is less affected by interaction with the tropopause and the stratopause.

## 5.2 Water Vapor Intrusion due to Tropopause Heating

As expected, warming of the tropical tropopause layer by radiative heating of volcanic debris facilitates the cross-tropopause troposphere-to-stratosphere transport of water vapor (Oltmans and Hofmann, 1995; Nedoluha et al., 1998; Joshi and Jones,
2009). The presence of extra water vapor in the stratosphere intensifies OH production and accelerates $SO_2$ oxidation to form sulfate particles (LeGrande et al., 2016).

For the 1s1-17km experiment, the stratospheric (i.e., above $100\,\mathrm{hPa}$) water vapor mass increases by about $30\,\mathrm{Mt}$ (Fig. 7a) at the equatorial belt. However, changes of water vapor above the tropopause do not affect volcanic cloud evolution much because the bulk of this water vapor is well below the altitude where the core of the volcanic cloud resides. Cross-tropopause water
transport decreases as injection height increases. For example, the 1s1-25km injection experiment shows no cross-tropopause water transport. In three weeks after the injection, the aerosol water associated with sulfate aerosols in the 1s1-17km experiment (Fig. 7c) is higher compared with other experiments, because in the 1s1-17km experiment more water vapor penetrates the stratosphere through the tropopause. However, the mass of $SO_4^{2-}$ in the 1s1-20km run continues to increase during August and September 1991 (Fig. 3b), and the associated aerosol water also increases to $3.5\,\mathrm{Mt}$ as shown in Fig. 7c. Little ice is
accumulated in the stratosphere in all experiments (Fig. 7b), since it is quickly removed by gravitational sedimentation. In the 1s1-20km experiment liquid water mass peaks at $3\,\mathrm{Mt}$; in the 1s1-17km - at 2Mt; and in 1s1-25km - at $1\,\mathrm{Mt}$ (Fig. 7d).





### 5.3 Volcanic Water Injection

Water vapor injected into the stratosphere with a volcanic plume could directly affect the initial evolution of a volcanic cloud since it is concentrated within it. Most of this water is brought by entrainment of tropospheric water in an explosive jet or
co-ignimbrite convective updrafts; nevertheless, the term "volcanic" water is used here. A wide range (75 Mt - 150 Mt) of volcanic water vapor injection for the Pinatubo eruption was reported (Joshi and Jones, 2009; Nedoluha et al., 1998). However, the amount of volcanic water retained in the stratosphere depends on the height of the injection. That is, almost all water vapor injected at a low temperature above the tropopause forms ice and quickly sediments (Stenchikov et al., 2021). A larger fraction of water vapor injected at higher altitudes, where stratospheric temperatures are higher, could remain in the stratosphere. To
test the sensitivity of volcanic clouds to the amount of volcanic water vapor, we conduct the 1w1 simulations injecting $SO_2$ and 15 Mt or 150 Mt of water vapor at 20 km and 25k km heights.

Figure 8 compares the time series of the equatorial SAODs, and changes in the globally integrated masses of sulfate and water species in the stratosphere (above 100 hPa) in the 1w1 experiments, with the simultaneous injection of $SO_2$ and 15 Mt or 150 Mt of volcanic water vapor at 20 km and 25 km with respect to corresponding 1s1 experiments (see Table 3). Water
species comprise water vapor, ice, and aerosol water. The aerosol water accumulates in sulfate and over ash particles.

The effect of volcanic water on the generation of the $SO_4^{2-}$ mass and SAOD is dependent on the amount of water vapor retained in the stratosphere. The sensitivity of SAOD and $SO_4^{2-}$ mass to the injected volcanic water vapor is higher in the 1w1-25km experiment compared to the 1s1-20km experiment (Fig. 8c,d). The increase in sulfate mass results from acceleration of $SO_2$ oxidation facilitated by the higher water vapor concentration (see Eq. R4). Water vapor emission in the 20 km injection
experiment has a weaker effect than in the 25 km injection experiment, because most of the water vapor injected at 20 km condenses and deposits from the stratosphere, since the temperature is lower at 20 km than at 25 km (Fig. 8i). Because more injected water remains in the stratosphere in the 1w1-25km experiment, its effect is more significant than in the 1w1-20km experiment.

### 5.4 Volcanic Ash Injection

In the va0 and va1 experiments, we inject 75 Mt of ash together with $SO_2$ and water vapor. The va1 experiment accounts for ash aging, but the va0 does not. In both experiments, we assume that the volcanic ash is initially hydrophobic. Therefore, we inject it into the insoluble (dry) accumulation and coarse modes (Fig. 2 and Fig. 9). In the va1 experiments volcanic ash ages quickly, populating the soluble (wet) modes (Fig. 9c,d), while ash particles in the va0 experiments remain in the dry modes (accumulation and coarse). In the va1 experiments, ash particles increase in size due to the aging and associated water and
$SO_4^{2-}$ uptake, which tends to transfer particles from the accumulation to the coarse mode.

In the va0 experiments, ash in the coarse mode (see Fig. 9b) sediments from the stratosphere in two days, but ash particles in the accumulation mode remain in the stratosphere for a week (Fig. 9a). In the va1 experiments, the ash mass in the wet modes increases quickly due to dry-to-wet particle conversion shown by the arrow in Fig. 9. The aging of ash particles slows the decrease of ash mass in both accumulation and coarse modes.





In the experiments with 25 km injection height, it takes longer for ash to reach the tropopause and leave the stratosphere in comparison to the 20m km experiment. For instance, after the first day of injection, 60 Mt of insoluble coarse mode ash mass remains in the stratosphere (not shown) for the va0 experiment with 25 km injection height compared to 1.7 Mt when ash is injected at 20 km (see Fig. 9b).

      Figure 10a shows the evolution of the stratospheric ash masses in the va0 and va1 experiments compared to the AVHRR

and HIRS/2 retrievals (Guo et al., 2004b). The mass of volcanic ash in va0 is smaller than that in va1. In the va1 experiment, the model ash mass is higher than in the AVHRR and HIRS/2 observations, while in the va0 experiment, the ash mass is underestimated when compared with observations. However, the uncertainties in the AVHRR derived ash mass are $\pm53\%$ (Gu et al., 2003) and $\pm85\%$ in HIRS observations (Yu and Rose, 2000).

      The injection of volcanic ash significantly increases stratospheric optical depth and $R_{eff}$ during the few days after injection.

This is shown in Fig. 11a,b which compares the time series of SAOD and the effective radius averaged within 20S - 20N latitude belt above 100 hPa in the 1s1-20km, 1w1-20km, va0-20km, and va1-20km experiments, with available observations. The AVHRR and scaled SAGE/ASAP SAODs are consistent for at least 4-5 months after the eruption. The CMIP6 SAOD appears to be half the size when compared with them. During the 4-5 months following the eruption, the simulated SAOD (Fig. 11a) is slightly larger than in observations but decreases more quickly when compared to observations later on in all

experiments except va0-20km. The va1 and va0 SAODs grow more rapidly during the first two months than in all experiments without ash injection. The effective radii in the va1 and va0 experiments spike to about 0.6-0.8 μm during the first week after the eruption, when a significant amount of ash is present in the volcanic cloud.

      Figure 11c,d shows the evolution of the $SO_4^{2-}$ mass in the coarse and accumulation modes in the same experiments integrated over the 20S - 20N latitude belt. In the va0 and va1 experiments, the stratospheric sulfate mass increases more rapidly than

in the 1w1 and 1s1 experiments. This is consistent with SAOD in Fig. 11a and with the more rapid depletion of $SO_2$ mass in Fig. 10b, which demonstrates a better agreement with $SO_2$ mass observations. We relate the faster $SO_2$ oxidation in the va0 and va1 experiments with the effect of heterogeneous reactions on ash particles, and more intensive volcanic cloud dispersion facilitated by ash radiative heating.

      Two months after the eruption, in the 1s1 and 1w1 experiments, the sulfate mass in accumulation and coarse modes reaches

maximums of 9Mt and 0.7 Mt, respectively. Thus, the sulfate formation rate increases in the va1 and va0 experiments compared to experiments without ash in both accumulation and coarse modes. The $SO_4^{2-}$ mass reaches the maximum two weeks earlier in experiments with ash than in experiments without ash (Fig. 11c,d).

      The aerosol water mass increases when sulfate mass increases, both for the accumulation and the coarse modes (Fig. 11e,f). A sulfate mass of 9Mt is associated with aerosol water mass of 3 Mt in the accumulation mode (Fig. 11c,e). This is consistent

with the 75% sulfuric acid solution assumed by Stenchikov et al. (1998). For the coarse mode, the aerosol water mass of 0.5 Mt is associated with roughly 0.8 Mt of sulfate (Fig. 11d,f). Both sulfate and wet ash particles accumulate aerosol water. In the long run however, due to the shorter lifetime of the ash particles, aerosol water is associated mainly with sulfate aerosols. Figure 11g,h show $SO_4^{2-}$ mass in the coarse and accumulation modes in the troposphere (integrated below 100 hPa) globally for the same experiments. Because of the rapid wet removal, little sulfate (not exceeding 0.4 Mt in each mode) is accumulated



in the troposphere both in accumulation and coarse modes. This is more than an order of magnitude less than the $SO_4^{2-}$ mass in the stratosphere. The tropospheric $SO_4^{2-}$ mass of volcanic origin comprises $SO_4^{2-}$ sedimented from the stratosphere. More sulfate mass sediments into the troposphere in the va0 and va1 experiments than in the 1s1 and 1w1 runs (Fig. 11e). This is because in the va1 and va0 experiments, the stratospheric sulfate mass is bigger than in the 1s1 and 1w1 experiments, as is $SO_4^{2-}$ deposition.

## 5.5   Ash Aging

Ash particles provide surface areas, enhancing the heterogeneous reactions and leading to significant changes in stratospheric chemistry (Danilin et al., 1999). Ash SAD is especially important in the first week after the eruption when few sulfate aerosols form. Fig. 12a-d show 20S-20N mean $SO_2$ mixing ratio and $SO_4^{2-}$ concentration as a function of time and height for the va0-20km and va1-20km experiments. Ash radiative heating causes lofting of the $SO_2$ plume by about 1 km per day in both

experiments, similar to that found in (Stenchikov et al., 2021), although ash in our simulations is more absorbing.

    In both cases, the $SO_4^{2-}$ cloud is below 35 km, but $SO_2$ reaches the stratopause. Therefore some $SO_2$ penetrates the mesosphere. This effect is more significant in the va0 experiment because of slower $SO_2$ oxidation compared with the va1 experiment. The enhanced mixing ratio of $SO_2$ in the mesosphere above 45 km was detected in ATMOS observations (Rinsland et al., 1995) and simulated in Brühl et al. (2015).

Figure 12e,g,i,k show the change in mixing ratio of $H_2SO_4$, aerosol water, OH, and $SO_4^{2-}$ concentrations in the va0-20km experiment with respect to 1w1-20km, in order to demonstrate the effect of ash injection. Fig. 12f,h,j,l show changes of the same characteristic, except in va1-20km with respect to va0-20km to demonstrate the effect of aging. If aging is turned on, $H_2SO_4$ condenses on volcanic ash, decreasing sulfuric acid concentration. At the same time, the presence of ash facilitates the heterogeneous reactions. The combined effect of ash aging and heterogeneous chemistry in our setting resulted in an increase

of the mass of sulphuric acid and sulfate by about 10%-20%, compared to those experiments without ash injections.

    Figure 13 shows the averaged over the tropical belt (20S-20N) shortwave and longwave heating rates caused by volcanic cloud for the va0-20km (left column) and va1-20km (right column) experiments, as function of time and height. The contour lines show the ash concentrations for the accumulation (top row) and coarse (bottom row) modes. There are two distinguished periods of shortwave heating in both experiments (Fig. 13a,b). The first period is just after the eruption, and the second is ten

days later. The first period is associated with ash solar absorption, and the second period with sulfate aerosol absorption. In both cases the shortwave heating by sulfate peaks at 25 km due to lofting of $SO_4^{2-}$ and ash plumes (see Fig. 12a,b). The average ash heating rates is about 0.4 $Kday^{-1}$ in experiment va1-20km, and 0.15 $Kday^{-1}$ - in experiment va0-20km (Fig. 13a,b). The shortwave heating caused by sulfate in the va1-20km experiment (Fig. 13a) is higher than in the va0-20km experiment (Fig. 13b).

The thermal absorption of the volcanic ash layer cools the top of the volcanic cloud during the first few days after the eruption. Still, absorption of upward IR radiation heats the bottom of the volcanic cloud. Heating caused by absorption of IR radiation by sulfate aerosols is seen in about ten days when enough $SO_4^{2-}$ is generated. Fig. 13d shows that in the va1-20km experiment, the longwave heating rate reaches 0.2 $Kday^{-1}$. We observe much weaker heating in the va0-20km experiment



(Fig. 13c). To summarize, we can say that ash aging significantly enhances the radiative effect of ash for a week after the
eruption.

## 5.6 Long-term climate response to volcanic forcing

In section 5.1 we showed that during the first six months after the eruption, the model demonstrates strong SAOD sensitivity to
the injection height. We also found that simulations with the volcanic emission of 17 Mt $SO_2$ at 20 km best fit the observations
during the six months after the eruption but overestimate SAODs. Here we further test the volcanic cloud evolution and
stratospheric temperature response for the entire post-eruption period of 2.5 years against observations. We take advantage of
the fact that the climate response provides another constrain to SAOD, since it defines stratospheric warming and tropospheric
cooling (Stenchikov et al., 1998; Kirchner et al., 1999). We also quantify the sensitivity of volcanic cloud evolution to the
amount of injected $SO_2$ considering the 12 Mt $SO_2$ emission at 20 km height.

Figure 14 compares the post-eruption evolution of SAODs (visible and near IR) and $SO_4^{2-}$ mass in the va1 experiments with
the 20km injection height and 17 Mt and 12 Mt $SO_2$ emission with the observations from CMIP6, AVHRR, SAGE/ASAP
(scaled visible and original NIR) SAODs for 2.5 years. The SAOD in the va1-25km experiment with 17 Mt $SO_2$ injection
SAOD overestimates the AVHRR and scaled SAGE/ASAP SAODs both in visible and near-IR (Fig. 14a-d) in July-August
1991. In the experiment with 12 Mt of emitted $SO_2$, the peak of SAOD is reduced and overestimates the observed SAOD
maximum only slightly in both visible and near IR. It is important to that the initial rate of development of visible and NIR
SAODs are similar in the model and observations both in the tropics and globally. It suggests the model on the stage when the
aerosol cloud is still confined in the tropics captures the $SO_2$ oxidation process and $SO_4^{2-}$ development quite well.

Starting from September 1991, the exaggerated speed of poleward transport of aerosols in the model causes a faster decrease
of SAOD in the simulations (both in the tropics and globally) than in the observations (Fig. 14e-h). This is because sulfur is
mainly deposited in the mid-latitude storm tracks through tropopause faults and in the polar regions in the downward branch
of B-D circulation (Gao et al., 2007), and the faster poleward aerosol transport makes both of these processes work more
effectively.

The CMIP6 visible SAOD is half the size during the first three months when compared to the scaled SAGE/ASAP and
AVHRR. This is primarily due to missing data in the original visible SAGE/ASAP data set.

We further evaluate the long-term model stratospheric temperature response to test the consistency of va1 simulations with
observations. Fig. 15 shows the temperature anomalies for the 1s1, va1-20km, va1-20km-12Mt, and MERRA2 reanalysis. The
left column of Fig.15 depicts the hovemoller diagrams of zonal mean temperature anomaly at 50 hPa, and the right column is
the temporal evolution of the global mean (70S-70N) temperature anomaly as a function of height or pressure. All experiments
in Fig. 15 resemble the spatial-temporal structure of the stratospheric temperature response well. They reproduce stratospheric
heating by the volcanic plume in the first year after the eruption, and the additional heating associated with the change of the
QBO phase in 1993. The simulations resemble the MERRA2 geographical temperate response and its vertical distribution well.

In the va1-20km experiment, the peak of temperature response is higher than in the va1-20km-12Mt experiment, reaching 4
K at 50 hPa. In the va1-20km-12Mt experiment, the temperature peak is half the size at about 2.5 K (Fig. 15a,c), which better





agrees with the MERRA2 temperature anomalies (Fig. 15e). Fig. 15b,d,f show a peak temperature anomaly at 30 hPa in the model simulations and the reanalysis. Again, the temperature response in the va1-20km-12Mt experiment fits the MERRA2

temperature anomalies better than the va1-20km experiment (Fig. 15b,d,f). Thus, reducing the injected SO$_2$ to 12Mt from 17 Mt shows a better agreement with the MERRA2 temperature response. It results in more realistic heating at 50 hPa in both tropics and subtropics. In the va1-20km-12Mt experiment, the SAOD is about 30% lower than in va1-20km. The lower SAOD causes weaker aerosol radiative heating and a less vigorous temperature response.

## 6 Conclusions

In this study we use the EMAC model with well-developed stratospheric chemistry (including heterogeneous chemistry) and detailed aerosol microphysics, to explore the evolution of the volcanic cloud from the 1991 Pinatubo eruption, the largest in the 20$^{th}$ century. We tested the model results with available observations of volcanic clouds and their radiative effect. We conducted ensemble simulations to study the impact of the injection height and its initial volume (one grid-box versus a latitude belt), as well as co-injection of water vapor, ash, and ash aging on the formation of the volcanic cloud.

The model simulations with 20 km injection height exhibit the best agreement of SO$_2$, SO$_4^{2-}$, and ash masses, with the AVHRR SAOD and SAGE/ASAP SAOD and SAD at different altitudes. In the 20 km injection experiments, the volcanic cloud is afterwards lifted to an altitude of 25 km by radiative heating, while in the experiments with volcanic materials injection at 25 km overshoots 30 km. The vertical distribution of SAOD and SAD in the observations and the model experiments with a 20 km injection height, show that the aerosol-cloud stabilizes in the middle of the stratosphere at 25 km. In the experiments with the

17 km and 25 km injection heights, the volcanic cloud interacts with the tropopause and the stratopause, respectively, causing aerosol mass to be lost too quickly. The stratospheric oxidation capacity and wind fields are different at different altitudes, strengthening the sensitivity to injection height. In the experiments with zonally uniform SO$_2$ injection in a latitude belt at a height of 25 km, the SAOD is significantly overestimated due to the higher oxidization rate.

Because of the coarse spatial resolution (T42L90), similar to other global models, EMAC simulates a too fast aerosol

poleward transport with a too quick escape of the volcanic materials from the tropical stratosphere. This process accelerates the loss of the aerosol mass to deposition at the poles and in the storm tracks.

The increase of water vapor in the stratosphere leads to an increase of the oxidization rate of SO$_2$ to SO$_4^{2-}$. The water vapor could be brought into the stratosphere by an eruptive jet, co-ignimbrite convection, and/or intruded through the tropopause heated by absorption of solar and IR radiation by volcanic debris. The cross-tropopause water vapor intrusion does not affect

the volcanic cloud evolution much, as most of the water penetrating through the tropopause accumulates below the volcanic cloud. The water vapor directly injected in the volcanic cloud in the 1s1-20km experiment increases the SO$_4^{2-}$ mass and SAOD by about 5%. The sensitivity of the SO$_4^{2-}$ mass to the amount of injected water in this experiment is low because most of the water vapor freezes and is quickly removed from the stratosphere in agreement with (Stenchikov et al., 2021). So the masses of remaining stratospheric water vapor in the 1w1-20km experiments with 15 Mt and 150 Mt of water vapor injections do

not differ much. A significant acceleration of SO$_2$ oxidation due to injection of water vapor (LeGrande et al., 2016) is only



reproduced in the experiments with the 25 km injection height, where temperature is higher than at 20 km, and a significant mass of injected water vapor is retained in the stratosphere.

The simulated mass of ash in our experiments is within the estimates of AVHRR and HIRS, but observations themselves are uncertain. Volcanic ash provides SAD for heterogeneous chemistry. This is most important during the few weeks after the eruption when ash is still abundant but sulfate aerosol is not yet developed. The simultaneous injection of water vapor and non-aging ash in the va0-20km experiment increases the maximum SAOD and $SO_4^{2-}$ mass by 10%.

In the va1-20km experiment, ash particles in the accumulation and coarse modes are entirely aged within a day after the injection. Aging increases the mass of ash particles. They continue up-taking water and $SO_4^{2-}$ molecules until removed by transport or sedimentation. The coarse ash particles deposit within a week, while it takes six months to reduce the mass of the ash accumulation mode from 1.2 Mt to 0.3 Mt. Overall, aging increases the SAOD by 20% and the $SO_4^{2-}$ mass by 10%. Aging doubles the radiative effect of ash both in SW and IR. The injections of volcanic water vapor and ash significantly accelerate the formation of the sulfate aerosols during the first two months after the eruption in the va1-20km and va0-20km experiments.

The simulated maximum SAOD and stratospheric temperature anomalies in the va1-20km-12Mt experiment with the 12 Mt $SO_2$ injection quite closely resemble the temperature anomalies obtained from the reanalysis both in latitude and height. The inclusion of volcanic ash adds to the radiative heating of the volcanic debris during the first week after the eruption in agreement with (Stenchikov et al., 2021), showing that the initial local heating results in lofting of the aerosol cloud. Our simulations show that the interactive calculations of OH and heterogeneous chemistry increase the volcanic cloud sensitivity to water vapor and ash injections, and have to be accounted for in simulations of volcanic impacts on climate and stratospheric chemistry.

*Code and data availability.* The EMAC code modifications, including all initialization data sets, and selected simulation results, are available at the KAUST repository site at https://repository.kaust.edu.sa/handle/10754/675509, DOI 10.25781/KAUST-0W317

*Author contributions.* MA performed the calculations and prepared all the figures. MA and GS wrote the manuscript. GS planned the analysis and calculations, led the discussion, and reviewed and improved the manuscript. AP, HT, and JL advised on EMAC modifications, discussed the results, reviewed and improved the manuscript.

*Competing interests.* The authors declare that they have no conflict of interest.

*Acknowledgements.* This research has been supported by the KAUST Competitive Research Grant (URF/1/2180-01-01) "Combined Radiative and Air Quality Effects of Anthropogenic Air Pollution and Dust over the Arabian Peninsula," and the KAUST Base Research Grant (BAS/1/1309-01-01). The authors thank the KAUST Supercomputing Laboratory for providing computer resources. We are thankful to



Christoph Brühl for valuable discussion and help in EMAC setting, and Linda and Mark Everett for proofreading this manuscript. HT ac-
635  knowledges funding from the Carl-Zeiss foundation and from the Deutsche Forschungsgemeinschaft (DFG, German Research Foundation)
– TRR 301 – Project-ID 428312742.



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



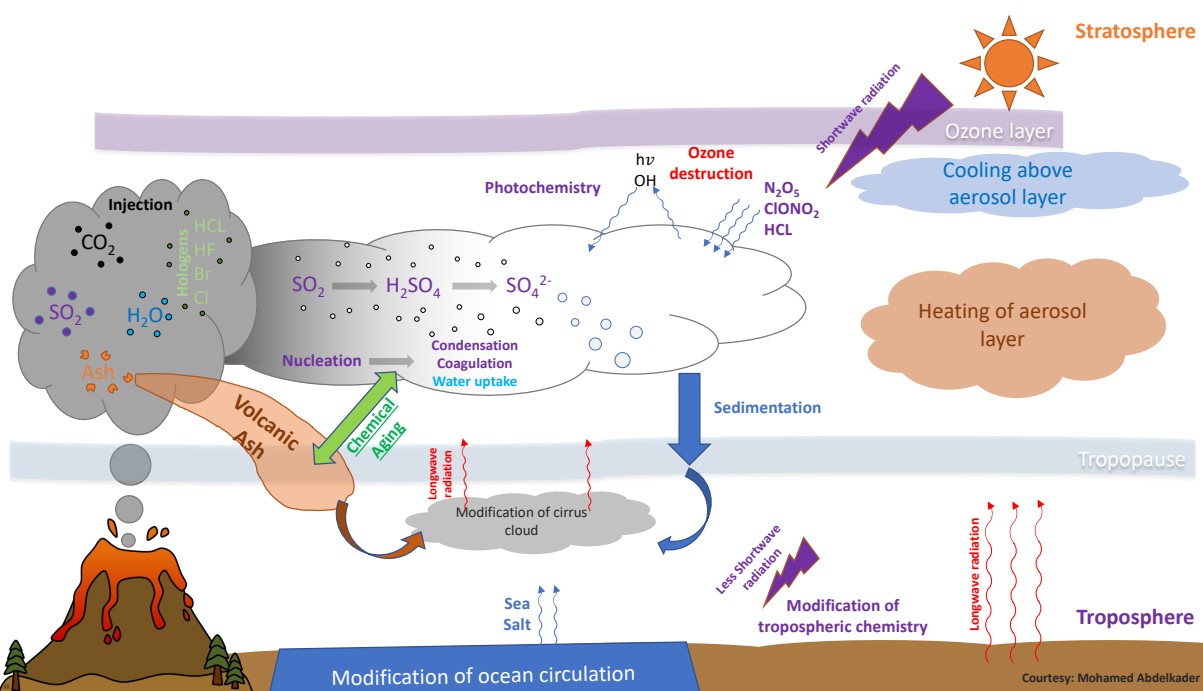

**Figure 1.** Schematic representation of volcanic eruption components in the Earth system.

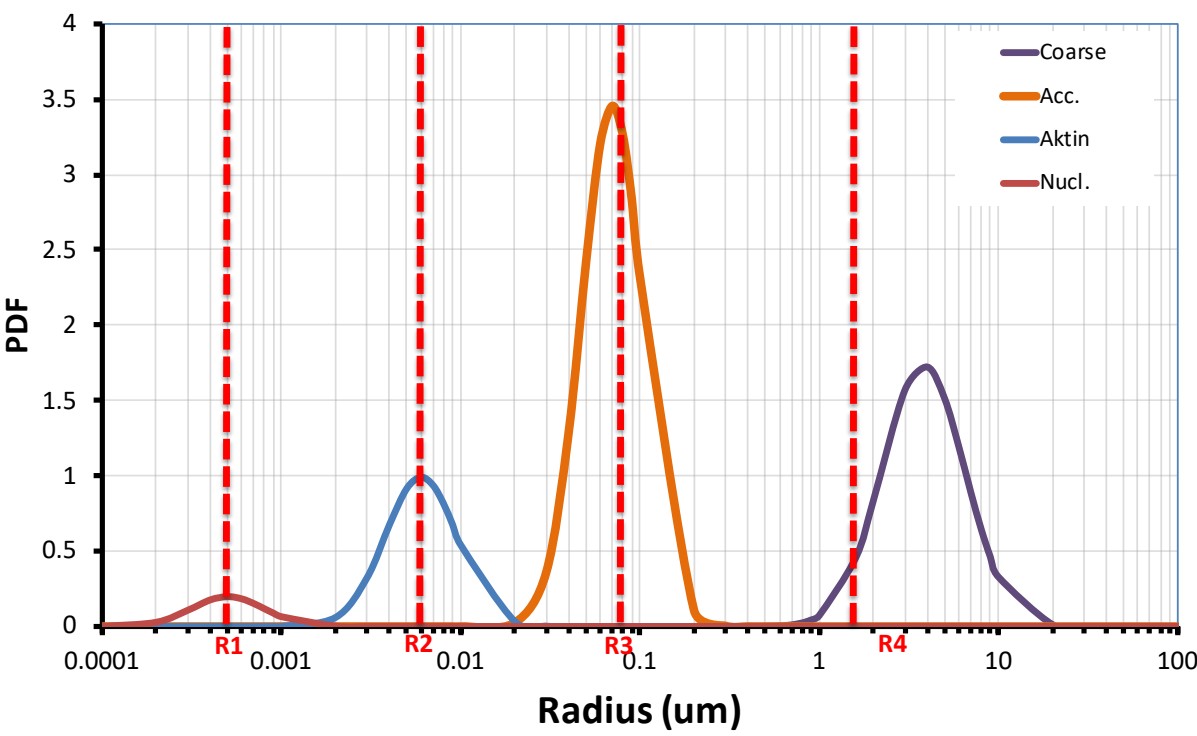

**Figure 2.** Schematic representation of the initial size distribution of aerosol modes in EMAC (nucleation, Aikten, accumulation and coarse). The threshold radii R1, R2, R3 and R4 control particle exchange between the modes. Initially 1.5 Mt of volcanic ash was injected in accumulation mode, and 73.5 Mt in coarse mode.



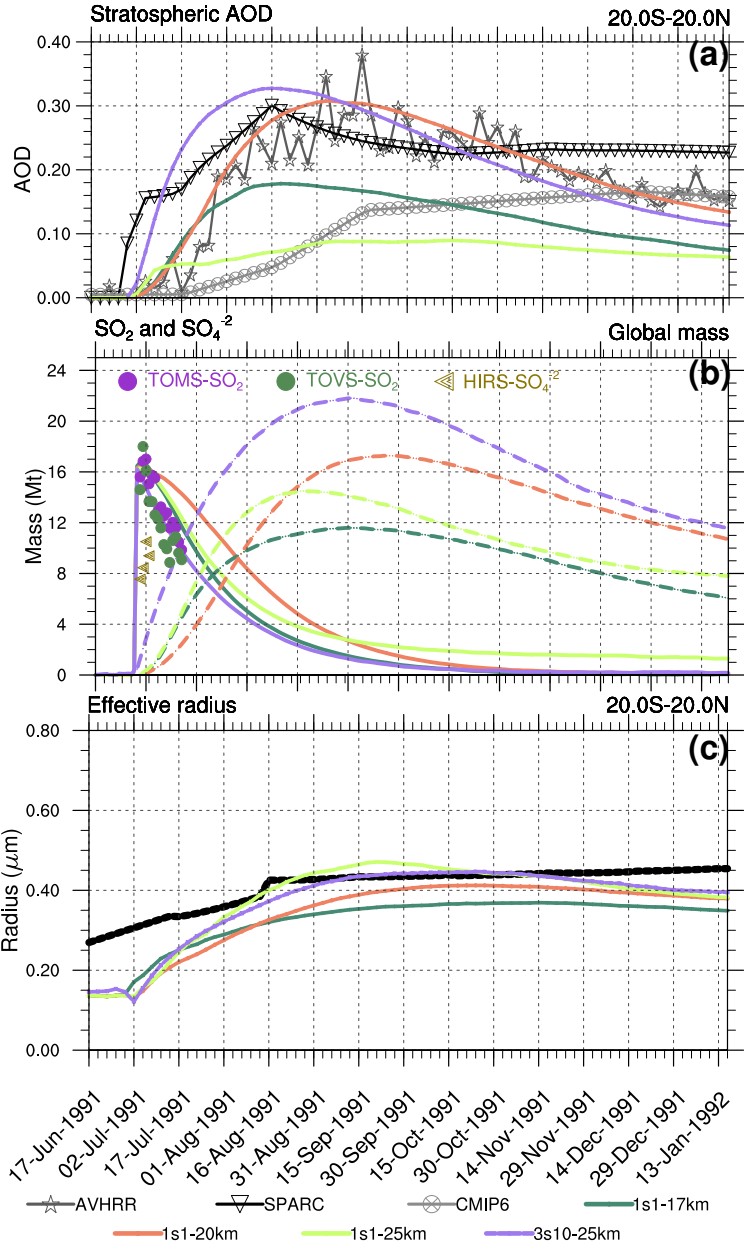

**Figure 3.** a) 20S-20N equatorial average visible SAOD from 1s1 experiment with the 17 km, 20 km and 25 km emission heights, AVHRR 0.630 μm, scaled visible SAGE/ASAP, and 0.525 μm CMIP6; b) $SO_2$ (solid lines) and $SO_4^{2-}$ (dashed lines) globally integrated masses calculated using output from 1s1 experiment with 17 km, 20 km and 25 km emission heights, the observed Guo et al. (2004b) $SO_2$ and $SO_4^{2-}$ masses are shown by markers; c) Equatorial average effective radius from 1s1 experiment with 17 km, 20 km and 25 km emission heights, and SAGE/ASAP retrievals (solid black).

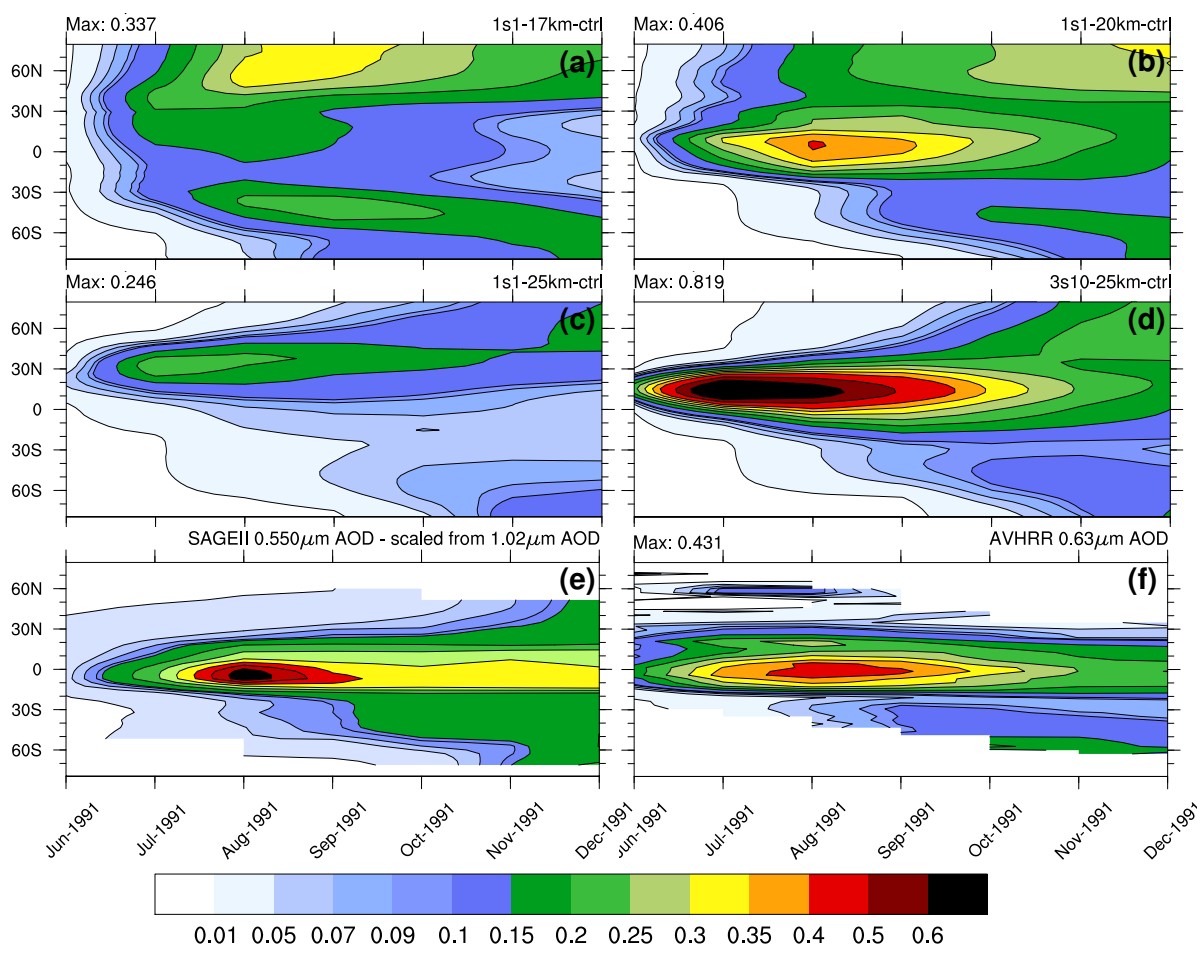

**Figure 4.** Zonally average visible SAOD as a function of latitude and time. a) 1s1-17km, b)1s1-20km, c) 1s1-25km, d) 3s10-25km, e) scaled visible SAGE/ASAP, f) 0.630 μm AVHRR.

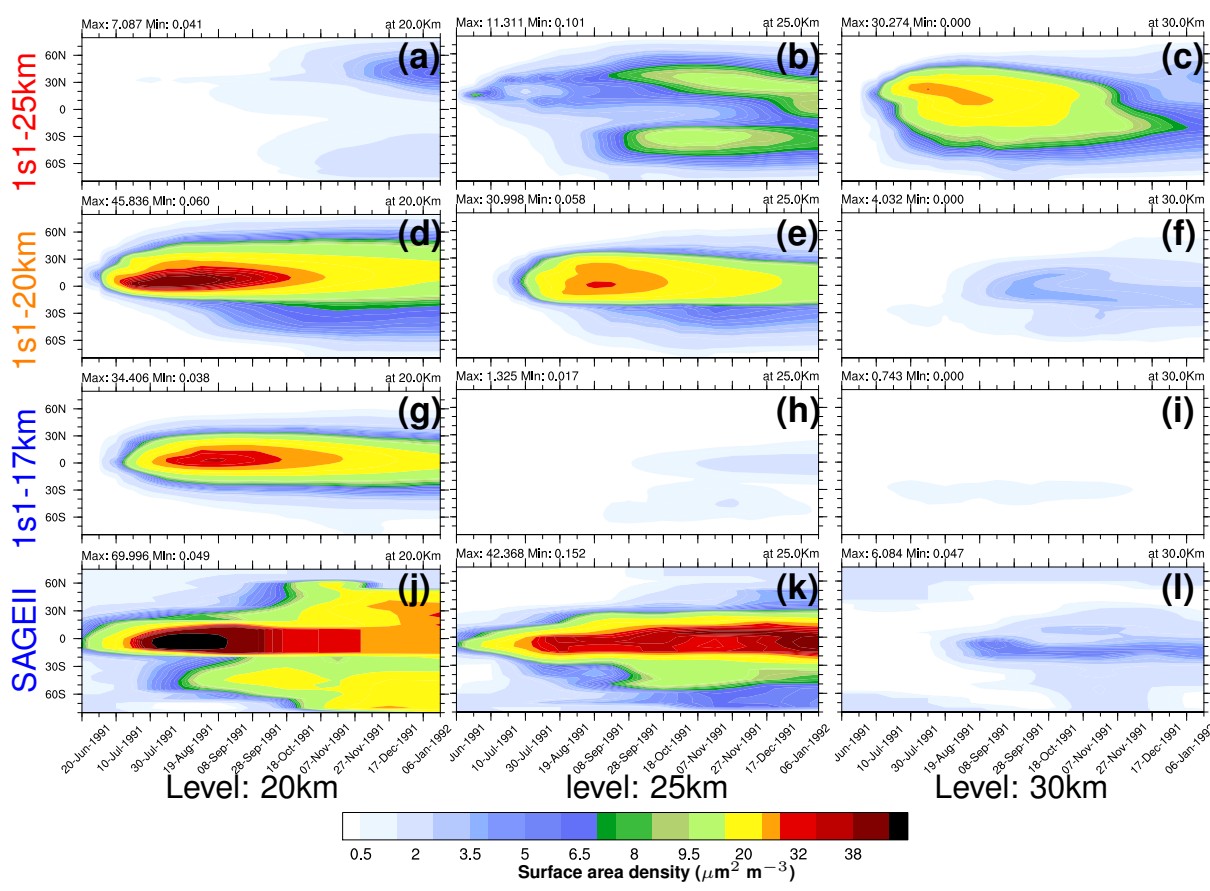

**Figure 5.** Zonally average Surface Area density (SAD, $\mu m^{-2} cm^{-3}$) as a function of latitude and time at 20 km (left, a,d,g,j), 25 km (middle, b,e,h,k), and 30 km (right, c,f,i,l). a-c) 1s1-25km, d-f) 1s1-20km, g-i) 1s1-17km, j-l) SAGE/ASAP



**Figure 6.** Perturbations (with respect to control) of 20S-20N equatorial average chemical constituents as a function of pressure (from 300 hPa to 1 hPa) and time in 1s1-17km (left, a,d,g,j,m,p), 1s1-20km (middle, b,e,h,k,n,q), and 1s1-25km (right, c,f,i,l,o,r). a-c) $SO_2$ (ppbv), d-f) $SO_4^{2-}$ (ngm$^{-3}$), g-i) OH (pptv), j-l) $H_2SO_4$ (pptv), m-o) $NO_x$ (ppbv), p-r) $NO_y$ (ppbv), $NO_x = (NO+NO_2)$, $NO_y = (NO_x + NO_3 + HNO_3 + 2*N_2O_5 + HONO + HNO_4 + ClONO_2 + BrONO_2)$

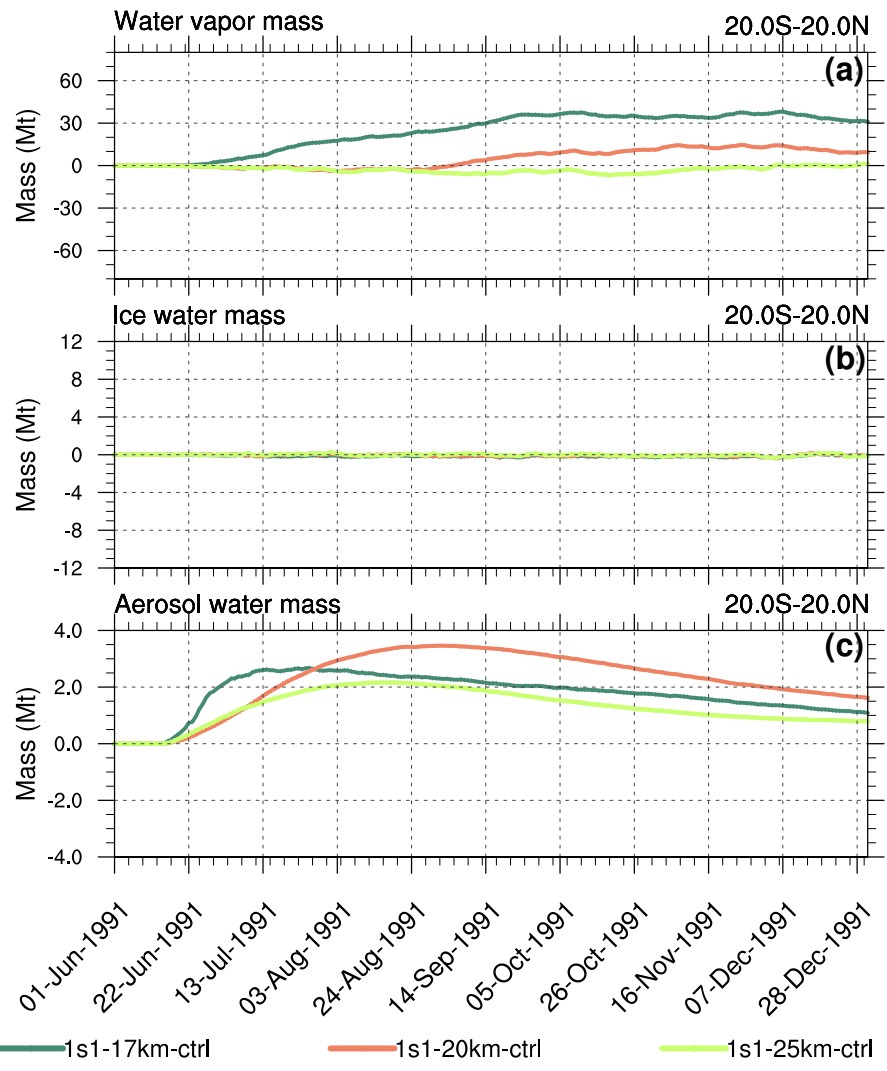

**Figure 7.** Perturbations (with respect to control) of 20S-20N stratospheric (above 100 hPa) integrated masses (Mt) in the 1s1 experiments with injection heights 17 km, 20 km and 25 km, as a function of time. a) water vapor, b) ice, c) aerosol water.

**Figure 8.** Perturbations (with respect to control) of 20S-20N stratospheric (above 100 hPa) SAOD and integrated masses (Mt) in the 1w1-20km and 1w1-25km experiments with respect to, correspondingly, 1s1-20km and 1s1-25km experiments as a function of time. Left column (a,c,e,g,f,k) - 1w1 experiments with 15 Mt of volcanic water vapor injection and right column (b,d,f,h,j,l) - 1w1 experiments with 150 Mt water vapor injection. a-b) SAOD, c-d) $SO_4^{2-}$, e-f) aerosol water, g-h) ice, i-j) water vapor.



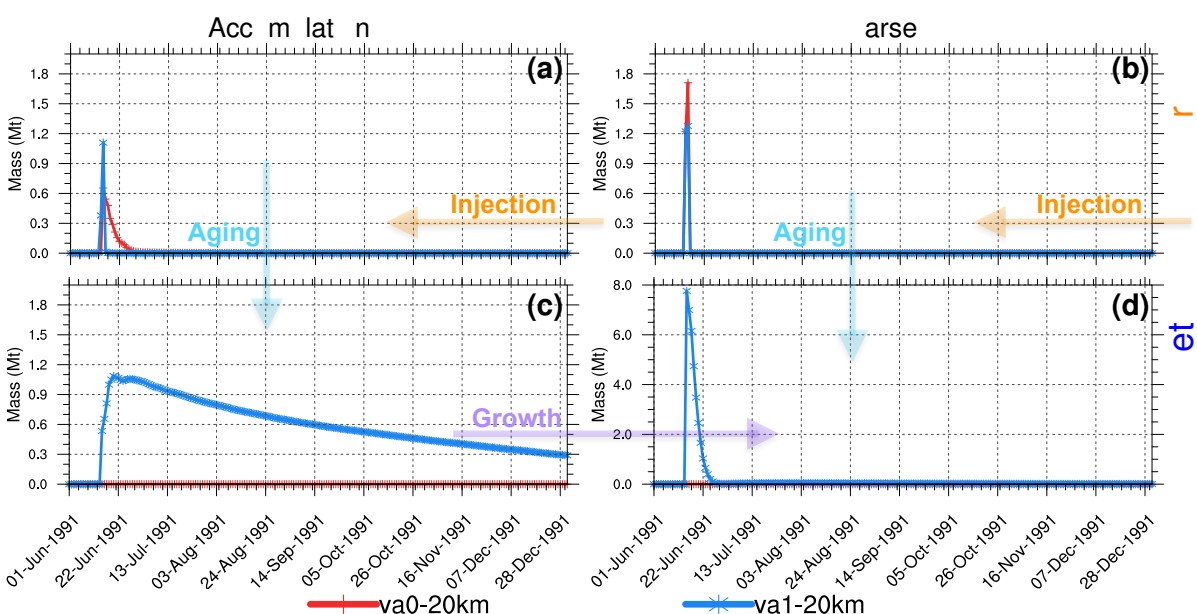

**Figure 9.** Globally integrated stratospheric volcanic ash mass (Mt) above 70 hPa as a function of time in the va0-20km and va1-20km experiments. a) Dry accumulation mode, b) Dry coarse mode, c) Wet accumulation mode, d) Wet coarse mode.



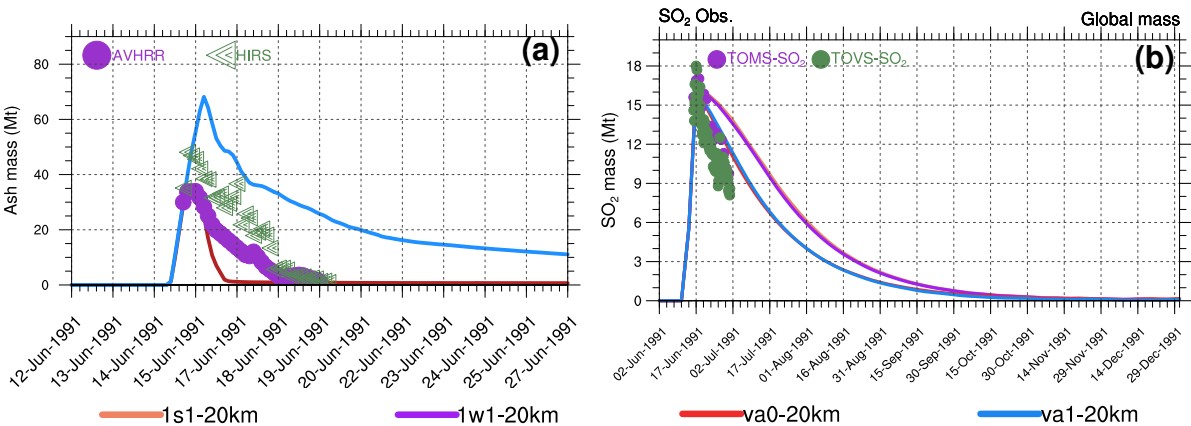

**Figure 10.** Globally integrated stratospheric masses (Mt) as a function of time. a) Volcanic ash in va0-20km, va1-20km, as well as in AVHRR, and HIRS retrievals Guo et al. (2004b), b) $SO_2$ in the 1s1-20km, 1w1-20km, va0-20km and va1-20km experiments, as well as in TOMS and TOVS observations.

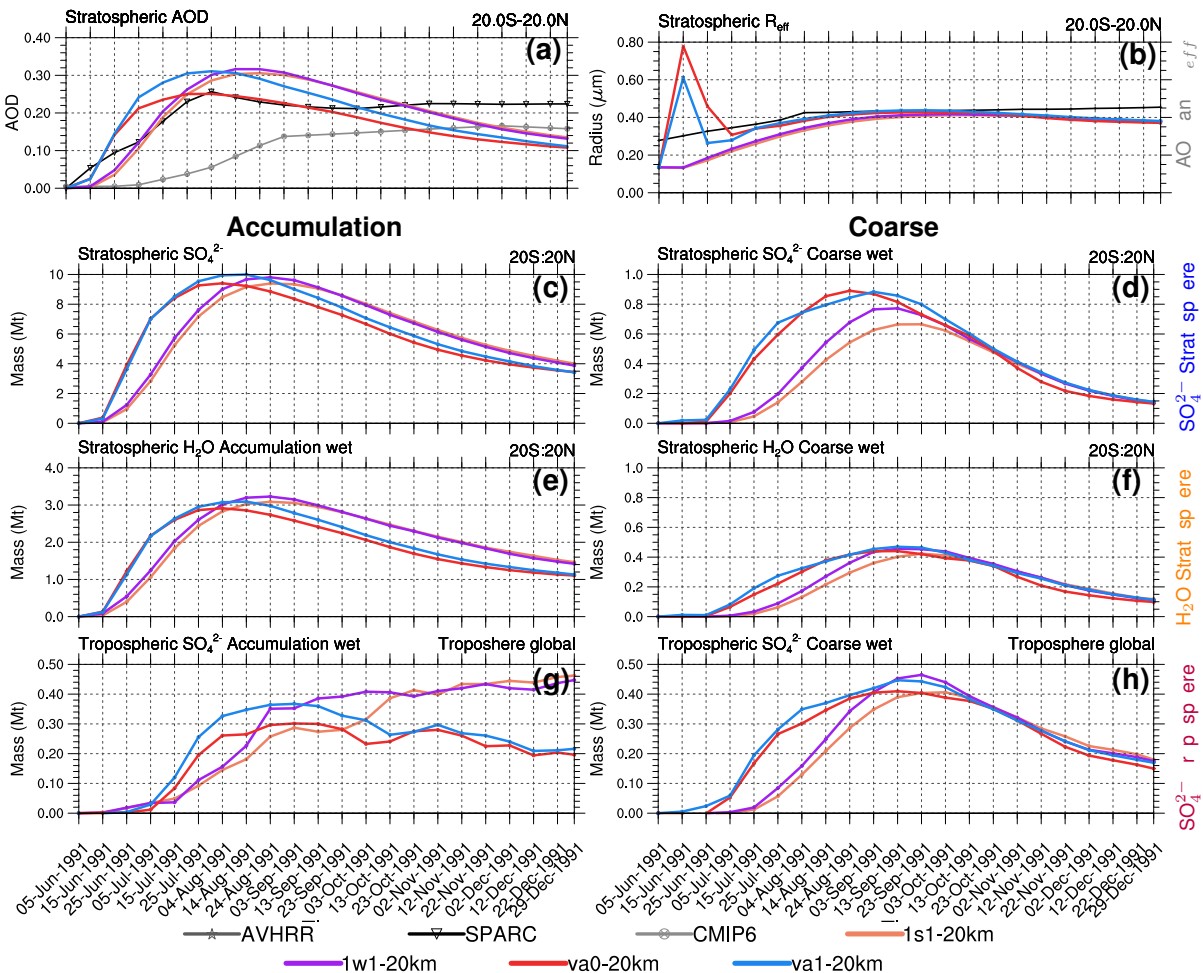

**Figure 11.** Visible SAOD, Aerosol efective radius, $R_{eff}$ above 100 hPa, and integrated masses (Mt) simulated in the 1s1-20km, 1w1-20km, va0-20km, va1-20km as a function of time. a) simulated, as well as observed AVHRR, scaled SAGE/ASAP, and CMIP6 20S-20N SAODs, b) Simulated stratospheric (above 100 hPa) $R_{eff}$, as well as observed in SAGE/ASAP in 20S-20N, c) Stratospheric $SO_4^{2-}$ in accumulation mode in 20S-20N, d) Stratospheric $SO_4^{2-}$ in coarse mode in 20S-20N, e) Stratospheric aerosol water in accumulation mode in 20S-20N, f) Stratospheric aerosol water in coarse mode in 20S-20N, e) Tropospheric (below 100 hPa) $SO_4^{2-}$ in accumulation mode integrated globally, f) Tropospheric $SO_4^{2-}$ in coarse mode integrated globally.



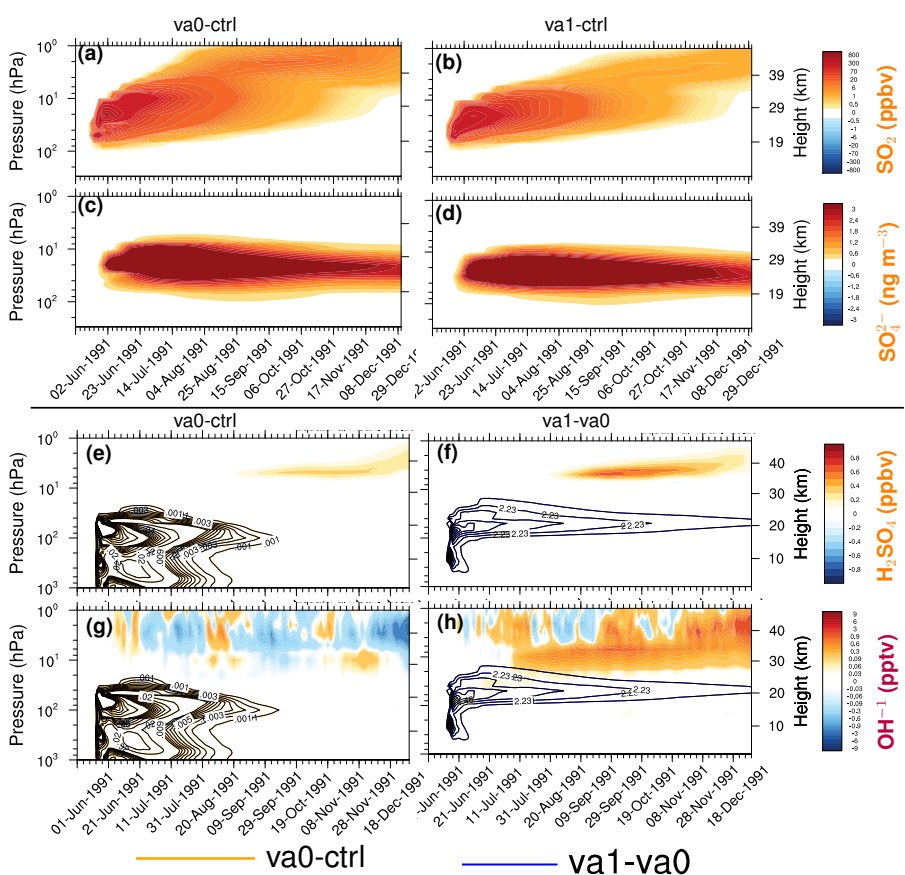

**Figure 12.** 20S-20N average perturbations of chemical constituents as a function of pressure (from 300 hPa to 1 hPa) and time in va0-20km and va1-20km experiments. a) $SO_2$ in va0-ctr (ppbv), b) $SO_2$ in va1-ctr (ppbv), c) $SO_4^{2-}$ in va0-ctr (ngm$^{-3}$), d) $SO_4^{2-}$ in va1-ctr (ngm$^{-3}$), e) $H_2SO_4$ in va0-1w1 (ppbv), f) $H_2SO_4$ in va1-va0 (ppbv), g) OH in va0-1w1 (pptv), h) OH in va1-va0 (pptv). The contour lines shows the accumulation mode ash mixing ratio (ppbv); orange contour lines for va0 and blue contour lines for va1-va0 in the (e-h) panels.



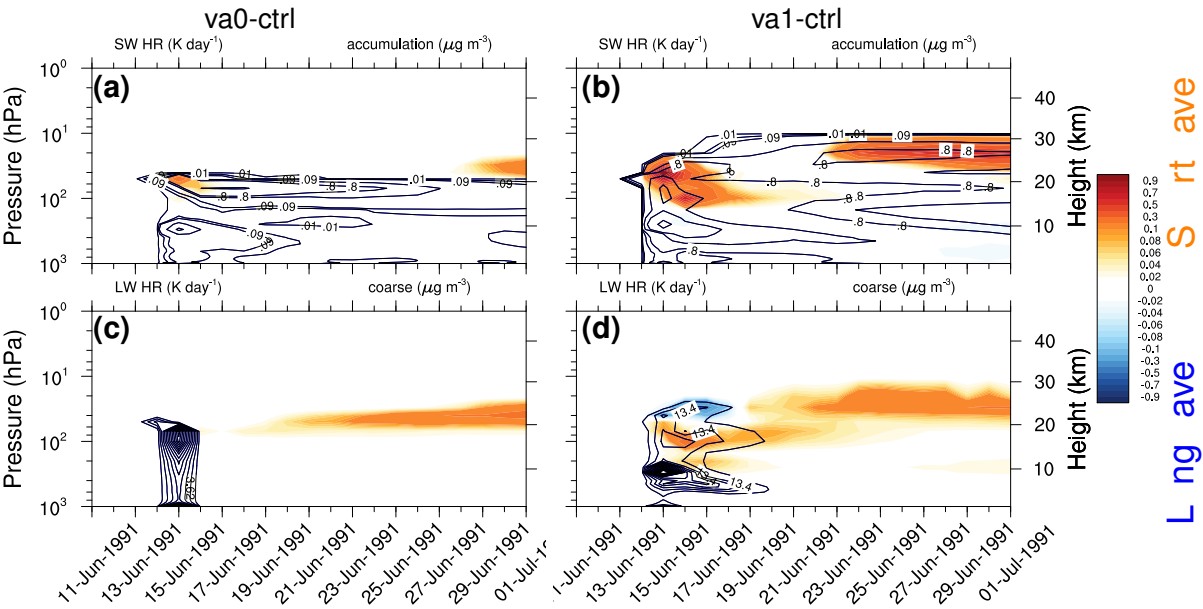

**Figure 13.** Averaged over the tropical belt (20S-20N) shortwave heating rate (K/day) shown as function of time and height overlaid by contours of volcanic ash mass concentration (accumulation mode) for a) va0-20km and b) va1-20km experiments, and longwave heating rate (K/day) for c) va0-20km, and d) va1-20km experiments both overlaid by contours of volcanic ash mass concentration (coarse mode). All heating rates are calculated by double call of the radiation routines.



**Figure 14.** Visible and NearIR SAODs in the va1-20km and va1-20km-12Mt experiments, as well as in AVHRR, scaled SAGE/ASAP, and CMIP6. a) Visible 20S-20N SAODs as function of time, b) NearIR 20S-20N SAODs as function of time, c) Visible globally averaged SAODs as function of time, d) NearIR globally averaged SAODs as function of time, e) Simulated visible zonally average SAOD in va1-20km as a function of time and latitude, f) Simulated NearIR zonally average SAOD in va1-20km as a function of time and latitude, g) SAGE/ASAP scaled visible zonally average SAOD as a function of time and latitude, h) SAGE/ASAP NearIR zonally average SAOD as a function of time and latitude.

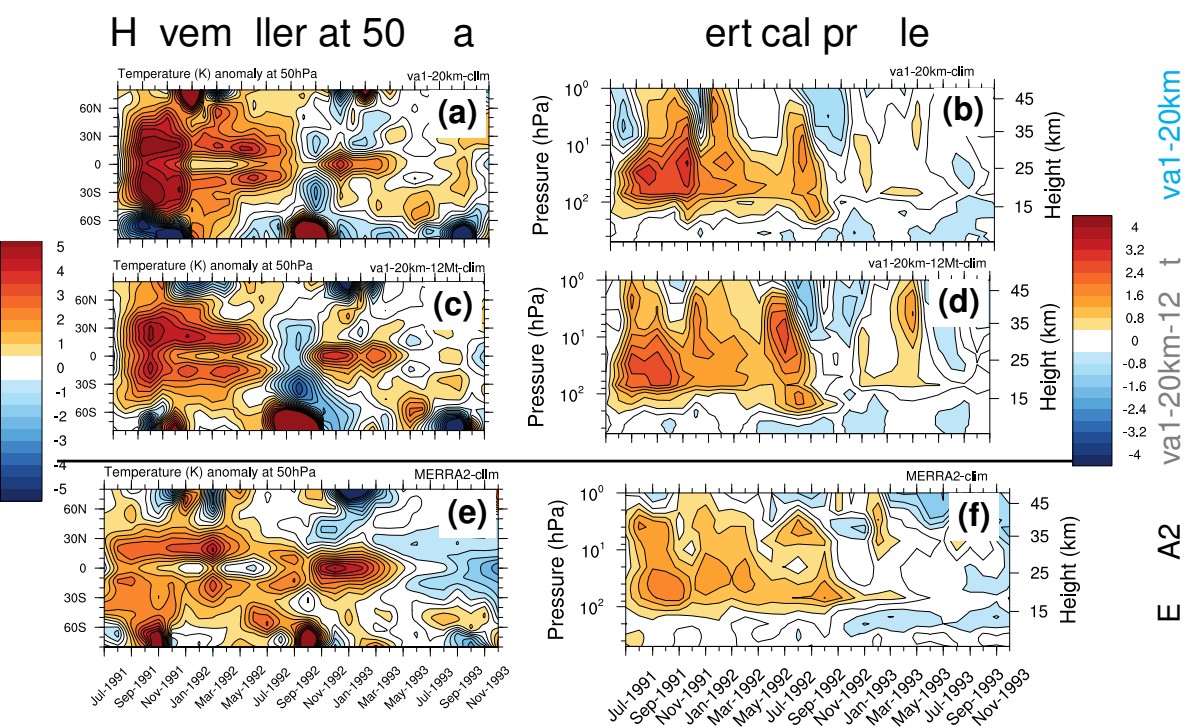

**Figure 15.** Atmospheric temperature anomalies (K) for the post-Pinatubo period with respect to the 1990-2000 climatology from the va1-20km (a,b), va1-20km-12Mt (c,d), and MERRA-2 reranalysis (e,f). The left column depicts zonally average anomalies at the 50 hPa pressure level as a function of time and latitude, and the right column depicts globally (70S-70N) averaged anomalies as a function of time and height/pressure.





**Table 1.** List or EMAC submodels used in this study. A complete list of all EMAC submodels can be found in Joeckel et al. (2010)

| Submode | Description | Reference |
|---|---|---|
| AEROPT | calculation of aerosol optical properties. | Klingmüller et al. (2014) |
| AIRSEA | air-sea exchange of trace gases | Pozzer et al. (2006) |
| CLOUD | ECHAM5 cloud scheme as MESSy submodel | Roeckner et al. (2006) |
| CONVECT | convection parameterisations | Tost et al. (2006b, 2010) |
| CVTRANS | convective tracer transport | Tost et al. (2006b) |
| DDEP | dry deposition of trace gases and aerosols | Kerkweg et al. (2006a) |
| GMXE | aerosol microphysics and gas aerosol partitioning | Pringle et al. (2010) |
| JVAL | photolysis rates | Landgraf and Crutzen (1998); Sander et al. (2014) |
| LNOX | production of $NO_x$ from lightning | Tost et al. (2007) |
| MECCA | atmospheric chemistry computations | Sander et al. (2011) |
| OFFEMIS | prescribed emissions of trace gases and aerosols | Kerkweg et al. (2006b) |
| ONEMIS | on-line calculated emissions of trace gases and aerosols | Kerkweg et al. (2006b) |
| RAD | ECHAM5 radiative transfer as EMAC submodel | Roeckner et al. (2006); Joeckel et al. (2006) |
| SCAV | scavenging and wet deposition of trace gases and aerosol | Tost et al. (2006a) |
| SEDI | sedimentation of aerosol particles | Kerkweg et al. (2006a) |
| TNUDGE | Newtonian relaxation of species as pseudo-emissions | Kerkweg et al. (2006b) |
| TROPOP | calculation of tropopause height | Joeckel et al. (2006) |



**Table 2.** Shortwave and longwave bands used in the raditaion transfere calculations

| No. | Shortwave (μm) | longwave(μm) |
|---|---|---|
| 1 | 0.25-0.69 (Visible) | 3.3,3.8 |
| 2 | 0.69-1.19 (NearIR) | 3.8-4.2 |
| 3 | 1.19-2.38 | 4.2-4.4 |
| 4 | 2.38-4.00 | 4.4-4.8 |
| 5 | | 4.8-5.6 |
| 6 | | 5.6-6.8 |
| 7 | | 6.8-7.2 |
| 8 | | 7.2-8.5 |
| 9 | | 8.5-9.3 |
| 10 | | 9.3-10.2 |
| 11 | | 10.2-12.2 |
| 12 | | 12.2-14.3 |
| 13 | | 14.3-15.9 |
| 14 | | 15.9-20.0 |
| 15 | | 20.0-40.0 |
| 16 | | 40.0-1000 |




**Table 3.** Description of experiments. The experiments are labeled according to the initial injection size and constituents of the injected plume. All experiment with "1x1" format represents injection in one grid box, 3s10 represents zonal injection with 10 grid points in latitude direction, the letter "s" denotes that only $SO_2$ in injected (dry injection), letter "w" denotes that $SO_2$ and water vapor and injected (wet injection), and va0 injection of volcanic ash with no aging and va1 is aging case. for the 1w1,va0, va1 experiments is injected with 15Mt and 150Mt of water vapor each has 5 ensemble members.

| Case name | $SO_2$ mass (Mt) | Water vapor mass(Mt) | Ash mass (Mt) | Injection height (km) | Number of ensembles | Emission volume[*] |
|---|---|---|---|---|---|---|
| ctrl | - | - | - | - | 5 | - |
| 1s1-17km | 17 | - | - | 17 | 5 | 1 grid box[*] |
| 1s1-20km | 17 | - | - | 20 | 5 | 1 grid box[*] |
| 1s1-25km | 17 | - | - | 25 | 5 | 1 grid box[*] |
| 1w1-20km | 17 | 150 or 15 | - | 20 | 5x2 | 1 grid box[*] |
| 1w1-25km | 17 | 150 or 15 | - | 25 | 5x2 | 1 grid box[*] |
| va0-20km | 17 | 150 or 15 | 75 | 20 | 5x2 | 1 grid box[*] |
| va0-25km | 17 | 150 or 15 | 75 | 25 | 5x2 | 1 grid box[*] |
| va1-20km | 17 | 150 or 15 | 75 | 20 | 5x2 | 1 grid box[*] |
| va1-25km | 17 | 150 or 15 | 75 | 25 | 5x2 | 1 grid box[*] |
| 3s10-25km | 17 | - | - | 25 | 5 | Zonal[**] |
| va1-20km-12Mt | 12 | 150 | 75 | 17 | 1 | 1 grid box[*] |

[*] 1 grid box - 280x280 $km^2$ with thickness of 1 km at 17km and 20km altitude and 0.5km at 25km altitude

[**] 10 grid box in latitude and 10 grid boxes in height ( 5km thickness)





**Table 4.** List of the studies that simulated interactive chemistry for Pinatubo case and the injected $SO_2$ height, maximum AOD and the time (in months) for the maximum AOD.

| Ref. | Altitude range (km) | Initial thickness (km) | Max. AOD | Time of Max. AOD (months) | $SO_2$ Mass (Mt) |
|---|---|---|---|---|---|
| Aquila et al. (2012) | 16-18 | 2 | 2 | 10 | 20 |
| English et al. (2013) | 15.1-28.5 | 13.4 | 0.24 | 7 | 20 |
| Banda et al. (2013) | 15-30 | 15 | 0.15 | 6 | 18.5 |
| Dhomse et al. (2014) | 19 - 27 | 8 | 0.35 | 2 | 10 |
| Bândă et al. (2015) | 17-21 | 4 | NA | 2 | 18.5 |
| Sheng et al. (2015) | 17-30 | 7-12 | NA | 3 | 14-20 |
| Mills et al. (2016) | 18-20 | 2 | 0.15 | 2 | 12 |