# Peer review of "The effect of ash, water vapor, and heterogeneous chemistry on the evolution of a Pinatubo-size volcanic cloud"

_Atmospheric Chemistry and Physics, 2022_

## Author Comment (AC1)

**The effect of ash, water vapor, and heterogeneous chemistry on the evolution of a Pinatubo-size volcanic cloud**
**Response to referee #1**

We thank the anonymous referee for the comments. The manuscript uses the EMAC global chemistry-climate model to study the interactions between sulfur dioxide, water vapor, and ash, accounting for ash aging in the 1991 Pinatubo volcanic cloud. EMAC employs an advanced description of gaseous and heterogeneous chemistry and aerosol microphysics. Generally, the manuscript extends the study of Stenchikov et al. (2021), where they employed the modified WRF-Chem model to study the initial (three months) development of the 1991 Pinatubo volcanic cloud with 25-km grid spacing forced by radiative feedbacks of SO2, volcanic ash, and sulfate aerosols. The current study covers longer time scales of three years following the Pinatubo eruption with a grid spacing of 280 km. Below we highlight the significant new aspects of the current study that differ it from (Stenchikov et al., 2021):

- EMAC explicitly accounts for ash aging
- EMAC explicitly accounts for the hygroscopic growth of sulfate/ash particles
- EMAC interactively calculates ozone

Despite the significant differences between EMAC and WRF-Chem, both models agree in many aspects. The stratospheric optical depth and the heating rates for the Pinatubo case are similar. The lofting of the volcanic cloud due to the heating by ash and later by sulfate particles are similar. Both models show that the volcanic cloud stabilizes at 24km when SO2 and ash are injected at 20km. In addition, both models show that the effect of the injected water vapor depends on the mass of water retained in the stratosphere—a significant amount of injected water freezes and sediments from the stratosphere as ice particles.

Here are our point-to-point replies to the referee's comments, which are in red, while our replies are in black.

1. The authors may consider shortening the paper. There are sentences that do not provide much additional information and can be removed. For example, lines 457-458 basically repeats the previous sentence.
- We removed Fig. 1 and the redundant sentences in L457-458.

2. In most sensitivity tests, volcanic material was injected in a relatively thin layer in the atmosphere. There is recent evidence that the plume height can be quite different for different parts of the plume (and not necessarily 20 km). Can the authors comment on how this may or may not affect the simulations and conclusions?
- We studied the sensitivity of the volcanic cloud to the injection height considering the injections of SO2 at 17km, 20km, and 25km. The wind field, temperature, and water vapor concentrations are different at different heights resulting in a different rate of SO4 formation. In the 17km experiment, more water vapor transported from the troposphere was available (Fig. 7 in the MS), resulting in a higher oxidation rate. Still, the plume mixes down to the troposphere through the tropopause so that the

maximum AOD=0.18. In the 20km experiment, the volcanic plume stabilized at 24km, where ozone concentration reaches maximum and more OH through ozone photolysis is available. The maximum AOD, in this case, reaches 0.3. In the 25km experiment, the plume moves above 30 km, and some volcanic materials are transported to the mesosphere.

3. Similarly, as the recent Tonga eruption showed, ash and SO2 could be separated during the initial stage of the eruption. Can the authors also comment on any potential impact on the simulations, if ash was indeed injected at a different height than SO2 for Pinatubo?
- Tonga eruptions did not inject much ash. At least it was not detected. In our case, we injected SO2 and ash at the same volume, but the separation of SO2 and ash starts already during the 24-hour emission stage so that by the end of the emission stage (16 June 1991) ash cloud is below the SO2 cloud. To answer your question, the decrease in the height of the ash plume with respect to the SO2 plume will decrease the velocity of the SO2 cloud lofting.

4. Introduction: Lines 57 and 87 seem to be redundant. Overall, the introduction is quite long and can be shorter.
- We removed redundant L87

5. Figure 1 is only mentioned in the passing in the text. Perhaps it is not completely necessary.
- We removed Figure 1 and L76-77 from the manuscript.

6. Section 3.1: I'm not entirely sure if R1-R5 need to be included in a research paper.
- We removed reactions R1-R5 and modified the text accordingly.

7. Line 223: Fig. 8 doesn't show refractive index.
- We referenced the correct Figure in the supplement.

8. Line 232: specify what RRTM is.
- We added the following sentence to L232: "a Rapid Radiative Transfer Model (RRTM)"

9. Line 233-234: It appears that IR absorption by SO2 was ignored? Would that have any significant effects on the plume transport?
- Stenchikov et al. (2021) showed that the SO2 heating rates in the stratosphere, both solar and terrestrial spectra, are two orders of magnitude smaller than heating rates generated by ash and one order of magnitude smaller than heating rates caused by sulfate aerosols. Osipov et al. (2020) demonstrated that the SO2 radiative effect becomes dominant for volcanic eruptions more significant than Toba, i.e., about 100-1000 times stronger than Pinatubo.

10. Lines 405-415: elaborate a bit more on how NOx and NOy are affected?

- Heterogeneous reactions on aerosols explain the repartitioning between NOx and the reactive nitrogen reservoir NOy. Here, the main pathway in this transformation is the oxidation of NOx to form N2O5 which interacts heterogeneously with water to form HNO3. N2O5 can also interact with halogens on the surface of aerosols (sulfate or ash), but we don't consider halogen injection in the current study. The heterogeneous reaction of N2O5 and water on the surface of aerosols effectively depletes NO2 from the active reaction system depending on Surface Area Density (Fig. 5 in the MS). During the daytime, HNO3 can photo-dissociate and release OH and NOx, while at night time, the formation of HNO3 is one-way via oxidation of NOx and N2O5. N2O5 can decompose back to NO3 and NO2 either photochemically or thermally, depending on the overhead column of ozone. With increasing altitude, temperature increases, and the rate of thermal decomposition increases. The limiting factors in the heterogeneous formation of HNO3 are that of NO2, O3, and SAD (Seinfeld and Pandis, 2006).

  Fig.6m-r in the MS shows a strong dependence of NOx and NOy on the injection height. For the 1s1-17km injection, the depletion of NOx (Fig. 6m) is lower than for 25km injection (Fig. 6o), while the production of NOy at 25km (Fig. 6r) injection is higher than for 17km injection (Fig. 6p). At higher altitude, the ozone concentration and SAD (Fig. 5c in MS) is higher, and hence the formation of HNO3 is enhanced for the 1s1-25km experiment (Fig. S9 in MS supplement). Although the change in NOy for 1s1 experiments at 17km,20km, and 25km injection is insignificant (Fig. S9b in MS supplement), the heterogenous transformation from N2O5 to HNO3 is efficient. The transformation is enhanced (Fig. S9b,c,d in the MS supplement) by the injection of ash particles due to the additional SAD and heating by ash and the associated stronger lofting of the volcanic plume.

11. Figure 8, 9, 11, 13, 15: missing letters from labels.

- We are sorry, we did not find any missing letters in the labels on those figures. Would you please clarify?

12. Figure 9: are the data points in the plot temporally averaged? The initial mass does not match with the injected amount.

- The data in Figure 9 are not temporarily averaged. We emit volcanic materials for 24 hours, and ash particles are deposited quickly. L478 now reads: "The difference in the ash mass between va0 and va1 on the first day resulted from the fast removal of the ash during the injection phase."

13. Figure 12 and lines 520-524: what is the mechanism for OH change between the cases with and without ash aging.

- Aging ash particles in the va1 experiments are coated with sulfate, making them less absorbing than pure ash particles in the va0 experiments. This effectively increases UV photolysis rates and facilitates OH production through ozone photolysis.

14. Conclusions - given the results here, can the authors make some comments on the Tonga eruption? For example, with the strong perturbation of water vapor in the stratosphere, do the authors expect any significant differences in terms of sulfate formation for Tonga?

- The recent Tonga eruption injected the bulk of volcanic materials at 35 km compared with 20 km for the 1991 Pinatubo eruption. At this altitude, stratospheric temperature and ozone concentration are higher than at 20 km. Therefore, the more injected water is retained in the stratosphere, and oxidation is mostly faster because of high ozone concentration and more intensive UV radiation. Tonga emitted little SO2, so it is unlikely it would deplete the stratospheric water vapor if it did not inject that huge mass (100 Mt) of water.

---

## Author Comment (AC2)

**The effect of ash, water vapor, and heterogeneous chemistry on the evolution of a Pinatubo-size volcanic cloud**

Response to referee #2

Review of "The effect of ash, water vapor, and heterogeneous chemistry on the evolution of a Pinatubo-size volcanic cloud" by Abdelkader et al. In this study, the effects of varying injection parameters and heterogeneous chemistry are modeled for the Pinatubo 1991 eruption. For this, the coupled chemistry-climate model EMAC is run with prescribed Sea Surface Temperature and nudged Quasi-Biennial Oscillation. Model experiments are compared with available limited observations and reanalysis data. The study is of scientific interest and includes novel aspects. It may be publishable after considering the following general and minor comments carefully.

We thank the Reviewer for the insightful comments. The manuscript uses the EMAC global chemistry-climate model developed at Max Planck Institute for Chemistry in Mainz, Germany, to discuss the interactions between the volcanic emitted sulfur dioxide, water vapor, and ash, accounting for ash aging in the 1991 Pinatubo volcanic cloud. EMAC employs an advanced description of the gaseous, water phase, heterogeneous chemistry, and aerosol microphysics.

Below are our point-by-point replies to the referee. The referee's comments are red, while our replies are black.

**General comments**

1. This study follows previous model work from some of the co-authors. In (Osipov et al., 2021; Stenchikov et al., 2021; Osipov et al., 2020), the effects of interactive SO2 and photolysis rates (next to volcanic ash) were simulated for the Toba super-eruption and the Pinatubo eruption using the EMAC and WRF models, respectively. Why were these effects not taken into account in this study as well? The ratio and consequences of the missing effects need to be explained and discussed. Also why choose a different EMAC model set-up as in Osipov et al 2020/2021 or is it the same one?

This is a misunderstanding. Stenchikov et al. (2021), Osipov et al. (2020, 2021), and the current study used different atmospheric models. Stenchikov et al. (2021) modified the regional WRF-Chem model to study the initial three months of volcanic debris evolution to evaluate the radiative effects of ash, SO2, SO4, and injected water. Osipov et al. (2020, 2021) employed GISS ModelE to calculate the impact of Toba supereruption. The current study uses the EMAC model to quantify the effects of heterogeneous chemistry, ash aging, and the injection height of volcanic debris on the long-term evolution of the volcanic cloud. The EMAC setting was optimized for this task.

Stenchikov et al. (2021) showed that the SO2 radiative heating in the Pinatubo-size cloud is significantly smaller than that of ash and sulfate aerosols.  Osipov et al. (2020) demonstrated that the SO2 radiative effect becomes dominant for volcanic eruptions more significant than Toba, i.e., about 100-1000 times stronger than Pinatubo. Therefore, we neglected the SO2 radiative effects in our calculations. The same is true for the impact of SO2 on photolysis rates. The novelty of the current study in comparison with our previous work (e.g., Stenchikov et al., 2021) is in implementing heterogeneous chemistry, ash aging, and interactive ozone chemistry. We also conducted long-term 2.5-year simulations to compare our results with the observations of the post-Pinatubo stratospheric temperature changes.

Despite the significant differences between EMAC in the current study and WRF-Chem (Stenchikov et al., 2021), both models agree in many aspects. The optical depth, the aerosol radiative heating, and the lofting of the volcanic cloud are similar. Both models show that the volcanic ash stabilized a few km above the injection level. These confirm that the chosen EMAC configuration is adequate and provides a reasonable background for the volcanic cloud's long-term chemical and microphysical development.

2. The authors run 5 ensemble members for each of their experiments. Which atmospheric initial conditions were chosen and how large is the spread among the different ensemble members?

We initialize EMAC from ECMWF Reanalysis, and each ensemble member is calculated using initial conditions taken from different years. We conduct one-year spin-up calculations for each ensemble run.

Following the reviewer's suggestion, we calculated the spread of the results between the ensemble members for each experiment and show one-sigma error bars in figures 2, 6, 8, and 13in the updated MS.   The "error bars" are reasonably small for all globally averaged quantities. However, the number of

runs in each ensemble was certainly not enough to study the high-latitude dynamic responses.

3.  Please give some background and discuss the variability of the SAOD response and its effects at northern high latitudes as observed and modeled for the Pinatubo eruption (Toohey et al., 2014).

    Stenchikov et al. (2006), Driscoll et al (2012), and  Charlton-Perez et al (2013) showed that the IPCC AR4 and AR5 have the problem of producing a stronger northern polar vortex in response to low-latitude volcanic eruptions. Conveying this signal to the surface is even more problematic. Polvani et al. (2019) concluded that the positive AO phase after the Pinatubo eruption is only by chance.

    Toohey et al. (2014) further elaborated on the planetary-wave-based mechanism of winter warming after large low-latitude eruptions. Bittner et al (2016a/b) and Azoulay et al. (2021) showed that a stronger eruption could more reliably force a positive phase of the AO. Recent research shows that detecting and attributing dynamic responses require large ensembles.

    Our model developed a spectacular winter warming in 1992/1993, much stronger than in the winter of 1991/1992, as in observations partly because of the westerly phase of QBO in 1992/1993, as discussed in (Stenchikov et al., 2004). See the figure below. But we have only five ensemble members in each experiment, which is nearly insufficient to get robust high-latitude dynamic responses.

    But the polar dynamic responses are not the primary goal of this paper that focuses on the chemical and microphysical development of the volcanic cloud. Discussion of the dynamic stratosphere-troposphere interaction would divert us from the main objective of this study. Therefore, we added the review of the recent literature on stratosphere-troposphere dynamic interaction to the introduction but decided not to include an expanded analysis of the winter warming phenomenon in this study.

[Figure]

4. Why was the latitudinal band 3s10-25km experiment chosen? The motivation for this is rather vague. There is a bunch of other model studies, f.e., Dhomse et al. (2014) and Mills et al. (2016) next to (Brühl et al., 2015).

We compare our results with (Bruehl et al., 2015) because they use the same model and obtained much higher optical depth for the same SO2 emissions.

Stenchikov et al. (2021) indicated that a fresh volcanic cloud's chemical/microphysical transformations and lofting are sensitive to the initial concentrations of volcanic debris, i.e., the volume where they are initially released. The zonal mean initial distribution of injected materials is important because it is associated with the lowest initial concentration of SO2 for a given emission mass of volcanic debris. Therefore, e.g., the vertical lofting for this case would be the weakest. In EMAC, we found a strong sensitivity of volcanic cloud development to the injection configuration. We conducted a thorough analysis of this sensitivity using the model that explicitly accounts for radiative heating from ash and sulfate and interactively calculates the height of the volcanic cloud. We modified the text to clarify these points.

5. Stratospheric temperature response: Here it would be rather helpful to show the results from the other experiments. Suddenly the 20 km 12 Mt injection SO2 scenario comes up as a best analogy, but what do the others experiments show? As the MERRA2 reanalysis is based on a model as well, what do observations show for Pinatubo (cf. Labitzke and McCormick, 1992)?

We have shown that the va1-20km experiment produces the most realistic (compared with observations) spatial distribution of SAOD. In other experiments, the volcanic cloud gets asymmetrically shifted to the North Pole or interacts with the tropopause or stratopause layers too vigorously. We do not see much sense in presenting detailed responses in the experiments with unrealistic spatial-temporal development of the volcanic cloud. Instead, we compare the long-term

stratospheric temperature response with observations to further test the "best" va1-20km experiment. This comparison provides another constraint to the magnitude of the SO2 injection, which we try to take advantage of. It is similar to how Kirchner et al. (1999) compared with (Labitzke and McCormick, 1992). But in our current paper, we do not have to filter out the QBO signal, as we account for the right phase of the QBO in the simulations.

MERRA2 assimilates the stratospheric temperature observations. The MERRA2 temperature fields are consistent with the observations reported by Labitzke and McCormick (1992). In (S-RIR, 2022), the MERRA2 stratospheric temperature anomalies caused by the 1991 Pinatubo injection resemble the observations well despite the absolute stratospheric temperature being slightly underestimated.

We corrected the text to clarify all these points.

6. Overall, it would be interesting to see some of the results (SO2, SO4, SAOD, R eff, and stratospheric temperature) for all experiments, which would certainly lengthen the manuscript. Thus, I leave it up to the authors to decide but I think it would be very helpful for a better understanding and model intercomparison.

   See response to comment #5

7. The abstract and conclusions need some overall take home messages i) on the overall study conclusion, and ii) from the set of model experiments: Which model experiment fits best with observations?

   Following the reviewer's suggestions, we completely rewrote the abstract.

**Minor comments: Abstract:**

8. The volcanic cloud interacts with tropopause and stratopause,? coupled with the ozone cycle.? This sentence needs to be revised (science and grammar).
9. Pls add an overall conclusion wrt to the SO2, ash and water vapour injection impacts.
10. Pls add an overall model vs observation conclusion. Which model experiment is the closest to the Pinatubo observations within the EMAC model world?

   See our response to comment #7.

**Introduction:**

11. Volcanic activity is a major natural cause of climate variation?? Pls correct as not all volcanic activity is climate relevant. You are referring to major explosive volcanic eruptions reaching stratospheric levels only.

We changed the sentence in L18 to read:

Strong explosive volcanic eruptions are the major natural cause of climate variability on both global and regional scales (Robock, 2000). They inject a mixture of $SO_2$, volcanic ash, water vapor, halogens, and other tracers into the lower stratosphere.

12. dacitic magma: Explain dacitic and relate it to your research work here.

We removed "dacitic" in L36

13. A positive phase of the Arctic Oscillation is not simulated by recent CMIP models. See the more recent studies by Driscoll et al (2012); Charlton-Perez et al (2013); Toohey et al (2014); Bittner et al (2016a/b), and following work. This statement has to be updated with more recent research work and model results.

Please see our responses to comment # 3. The text is modified accordingly.

14. Line 36: Over which time period erupted Pinatubo?

We changed the sentence in L36 to read:
According to observations, three main volcanic explosions on 15 June 1991 spread volcanic ash and gases over an area of 300,000 $km_2$

15. and has been neglected in many previous studies (Niemeier et al 2009; Oman et al 2006).? Pls cite also more recent papers here.

We added a few more references and modified L47 to read:
The online calculation of OH is essential to correctly reproduce the dynamics of sulfate aerosol mass (Clyne et al., 2021; Stenchikov, 2021), and this has been

neglected in many previous studies (Marshall et al., 2018; Niemeier et al., 2009; Oman et al., 2006).

16. From line 49 onwards:
Pls clarify and disentangle observational versus model studies here. Right now, the paragraph mixes both although having quite different reasons for the spread and uncertainties.

In L49-L60 we discuss the recent modeling studies that calculated SO2 to SO4 oxidation and aerosol microphysics differently, resulting in large uncertainty in injected $SO_2$ mass required to generate the observed SAODs and the climate effect.

We modified the text in L51 to read:

Therefore, different Pinatubo modeling studies report a wide …

10. Timmreck et al (2018) gives an uncertainty of 10-20 Mt SO2 injection into the stratosphere for the Pinatubo eruption based on available observations and model work, which should be referred to here. Then the details before can be shortened.

We added the reference to (Timmreck et al., 2018)

11. Line 76-77: Fig. 1 is nice to have but you can also just refer to McCormick et al (1995); Robock (2000); Timmreck (2012); and Zhu et al 2020. There is nothing new you add here, or? Next, there are also processes displayed you do not address or mention (c.f. ocean circulation and biogeochemistry).

Thanks, we have removed Figure 1 from the manuscript.

12. Line 95 >: The difference to Stenchikov et al. (2021) is mentioned partly, but it still lacks that SO2 heating is not included next to online photolysis rates of volcanic aerosols in your study. Pls try to explain what you do in contrast to

Stenchikov et al. (2021) and Osipov et al. (2020, 2021) and why. This list is not complete yet.

Please see the response to comment #1

We modified L95 to read:

In addition to processes considered in Stenchikov et al. (2021), we explicitly calculate ash chemical aging, stratospheric ozone chemistry, and aerosol microphysical processes, accounting for the hygroscopic growth of sulfate/ash particles. However, we do not account for the heating by SO2 because for the Pinatubo case, it is much weaker than radiative heating from ash and sulfate aerosols (Osipov et al., 2021; Stenchikov et al., 2021; Osipov et al., 2020)

**2.3 Data**

13. How good is the MERRA2 assimilation product for the Pinatubo? Pls check the new S-RIP 2022 report. Pls compare with observations f.e Labitzke and McCormick, 1992.

    See our response to comment #5

14. Line 196: ? sulfate represents by the soluble mode? grammar correct?

    We modified the text in L196 to read:

    Sulfate particles are represented by soluble modes, while ash is initially considered insoluble.

**2.4 Model**

15. Line 164-167: I assume you also take into account natural and anthropogenic surface halogen emissions as background (such as CHBr3, CH2Br2, CH3Br, CFCs, halons)?

    Yes, We accounted for the background emissions of CFS's (CFCl3, CF2Cl, CH3CCl3, CCl4): HCFC: CH3Cl, CH3Br, Halons (CF2ClBr, CF3Br)

We modified the text in L165 to read:

We also account for the background emission of CFC's, halogens and Halons.

16. 3.4 Section: Pls clarify
-AEROPT: EMAC module?
-RAD: EMAC module?
-Fouquart and Bonnel (1980) part of EMAC?

-RRTM part of EMAC?

Table 1 lists the EMAC submodels used in this study, providing the corresponding explanations. "AEROPT and RAD" are the EMAC submodels (L226 and L230). "Fouquart and Bonnel" is the shortwave radiation scheme, and "RRTM" is the longwave radiation scheme in the "RAD" sub-model.

17. SO2 is not radiative active in this (EMAC) model study but it is included in EMAC used by Osipov et al 2020 and 2021, why not here? Pls explain the ratio and the effects of neglecting it.

See our response to comment #1

**2.5 Experimental Setup**

18. Line 251: Why choosing different injecting heights? This is not really motivated and explained in the introduction.

We modified the text in L86-95 to read:
Different modeling studies assume different injection heights. The results show that the oxidation rate of SO2 depends strongly on the injection height according to the availability of water vapor and OH radicals.

19. Line 256: 3s10-25km: So the injection layer is 22.5-27.5 km or ??

Yes.

20. Line 265: ?Based on different atmospheric initial conditions? Which are?

See our response to comment #2

**2.6 Results**

21. Line 296: ?The cloud height is essential?? Do you mean injection height? This whole sentence needs an overall rewording to make scientifically sense.

We meant the actual height where the cloud resides. This height changes over time and depends on the injection height. The actual height of the cloud defines the physical (wind, temperature) and chemical (ozone, water vapor, photolysis rates) environment that affects the transport and chemical evolution of the cloud. We clarified the text in L296.

22. Line 301: ?lofting driven by radiative heating of volcanic debris? So what is the effect of the missing SO2 radiative heating in your results? (see also Osipov et al 2020/2021; Stenchikov et al 2021)

See our response to comment #1

23. Why not continuing with model experiment 3s10-25km if it shows such a good comparison with observations? The ratio for this is missing.

In 3s10-25km experiment, the optical depth is overestimated. The conversion rate of SO2 to SO4 was increased for the wrong reason. See the discussion in section #5.1.3

24. Line 385: Stenchikov et al 2021 and Osipov et al 2020&2021 included online photolysis rates (of volcanic aerosols) in addition in contrast to your study here, nor?

See our response to comment #1

25. Section 5.1.5 and Figure 6:
Can you show O3 as well which would be interesting to see and to understand and interpret the stratospheric temperature response in Fig. 15?

The ozone panel was added in what is now Fig. 5. The ozone content changed by less than 5%. Therefore, except in polar regions, its effect on temperature response is negligible.

26. Section 5.6:
Pls compare also with observations cf. Labitzke and McCormick (1992).

Yes, in section 5.6, we explicitly discussed the comparison of simulated stratospheric warming with the observations of Labitzke and McCormick (1992).

**2.7 Conclusions**

27. Line 594-596: ?Because of the coarse resolution?similar to other global models?too fast aerosol poleward transport? ? This statement comes as a surprise. Can you pls elaborate a bit more on this and give references to it: Toohey et al (2014) simulates the effects of different Pinatubo aerosol forcing fields in MPI-ESM based on observations and MAECHAM5HAM model simulations (for 17 Mt SO2 injections representing different states of the NH polar vortex and thus aerosol transport and SAOD at high latitudes).

It is a known model deficiency. We discussed it, e.g., in (Oman et al., 2006). It is seen in Fig. 14 (e-h) in this manuscript. We expanded the discussion of Fig. 14 to clarify this point.

**2.8 Figures**

28. -The figures in the pdf file seem to have some problems. At same pages, letters are missing cf. Page 38 Y-axis labels on the right side, and Fig. 9 titles, etc.
Fixed

29. -Numbers at the legends are often unreadable cf. Fig. 6. This has to be checked and revised for all figures.
Fixed

30. -Figure captions need to explain the shown figures, which is often not the case, f.e. SPARC in Fig. 3 is missing etc.

We modified the figure caption to read:

SAGE/ASAP (Stratospheric Processes and its Role in Climate published in the Assessment of Stratospheric Aerosol Properties), f) 0.630 µm AVHRR (Advanced Very High-Resolution Radiometer).

31. Fig. 4 and elsewhere: Pls show meridional sections from 90N to 90S.

We could, but we focus here on the initial development of the cloud and want to better resolve the processes in the low latitudes

32. Fig. 5: SAGEII vs SAGE/ASAP ?

Thanks, we changed SAGEII to SAGE/ASAP in the figure caption.

33. Fig. 6: Ozone should be shown here as well.

Added

34. Fig. 8 and elsewhere: AOD, AO, vs SAOD is written, pls homogenize.

Fixed

**References**

1. Azoulay, A., Schmidt, H., & Timmreck, C. (2021). The Arctic polar vortex response to volcanic forcing of different strengths. Journal of Geophysical

Research: Atmospheres, 126, e2020JD034450.
https://doi.org/10.1029/2020JD034450

2. Bittner, M. (2015). On the discrepancy between observed and simulated dynamical responses of Northern Hemisphere winter climate to large tropical volcanic eruptions. PhD Thesis, Universität Hamburg, Hamburg. doi:10.17617/2.2239264.

3. Bittner, M., Schmidt, H., Timmreck, C., & Sienz, F. (2016). Using a large ensemble of simulations to assess the Northern Hemisphere stratospheric dynamical response to tropical volcanic eruptions and its uncertainty: LARGE ENSEMBLES FOR VOLCANIC ERUPTIONS. Geophysical Research Letters, 43(17), 9324–9332. https://doi.org/10.1002/2016GL070587

4. Bru˙hl, C., Lelieveld, J., Tost, H., Ho˙pfner, M., and Glatthor, N. (2015). Stratospheric sulfur and its implications for radiative forcing simulated by the chemistry climate model EMAC. *Journal of Geophysical Research: Atmospheres*, 120(5):2103–2118.

5. Charlton-Perez, A. J., Baldwin, M. P., Birner, T., Black, R. X., Butler, A. H., Calvo, N., et al. (2013). On the lack of stratospheric dynamical variability in low-top versions of the CMIP5 models: STRATOSPHERE IN CMIP5 MODELS. Journal of Geophysical Research: Atmospheres, 118(6), 2494–2505. https://doi.org/10.1002/jgrd.50125

6. Dhomse, S. S., Emmerson, K. M., Mann, G. W., Bellouin, N., Carslaw, K. S., Chipperfield, M. P., Hommel, R., Abraham, N. L., Telford, P., Braesicke, P., Dalvi, M., Johnson, C. E., O'Connor, F., Morgenstern, O., Pyle, J. A., Deshler, T., Zawodny, J. M., and Thomason, L. W. (2014). Aerosol microphysics simulations of the Mt. Pinatubo eruption with the UM-UKCA composition-climate model. *Atmospheric Chemistry and Physics*, 14(20):11221–11246. WOS:000344165800017.

7. Driscoll, S., Bozzo, A., Gray, L. J., Robock, A., and Stenchikov, G. (2012), Coupled Model Intercomparison Project 5 (CMIP5) simulations of climate following volcanic eruptions, J. Geophys. Res., 117, D17105, doi:10.1029/2012JD017607.

8. Kirchner, I., Stenchikov, G. L., Graf, H.-F., Robock, A., & Antuña, J. C. (1999). Climate model simulation of winter warming and summer cooling following the 1991 Mount Pinatubo volcanic eruption. Journal of Geophysical Research: Atmospheres, 104(D16), 19039–19055. https://doi.org/10.1029/1999JD900213

9. Labitzke, K., & McCormick, M. P. (1992). Stratospheric temperature increases due to Pinatubo aerosols. Geophysical Research Letters, 19(2), 207–210. https://doi.org/10.1029/91GL02940

10. LeGrande, A. N., Tsigaridis, K., and Bauer, S. E. (2016). Role of atmospheric chemistry in the climate impacts of stratospheric volcanic injections. *Nature Geoscience*, 9(9):652–655.

11. Mills, M. J., Schmidt, A., Easter, R., Solomon, S., Kinnison, D. E., Ghan, S. J., Neely, R. R., Marsh, D. R., Conley, A., Bardeen, C. G., and Gettelman, A. (2016). Global volcanic aerosol properties derived from emissions, 1990–2014, using CESM1(WACCM). *Journal of Geophysical Research: Atmospheres*, 121(5):2015JD024290.

12. Osipov, S., Stenchikov, G., Tsigaridis, K., LeGrande, A. N., and Bauer, S. E. (2020). The Role of the SO Radiative Effect in Sustaining the Volcanic Winter and Soothing the Toba Impact on Climate. *Journal of Geophysical Research: Atmospheres*, 125(2).

13. Osipov, S., Stenchikov, G., Tsigaridis, K., LeGrande, A. N., Bauer, S. E., Fnais, M., and Lelieveld, J. (2021). The Toba supervolcano eruption caused severe tropical stratospheric ozone depletion. *Communications Earth & Environment*, 2(1):71.

14. Polvani, L. M., Banerjee, A., & Schmidt, A. (2019). Northern Hemisphere continental winter warming following the 1991 Mt. Pinatubo eruption: reconciling models and observations. Atmospheric Chemistry and Physics, 19(9), 6351–6366. https://doi.org/10.5194/acp-19-6351-2019

15. Stenchikov, G., Hamilton, K., Robock, A., Ramaswamy, V., & Schwarzkopf, M. D. (2004). Arctic oscillation response to the 1991 Pinatubo eruption in the SKYHI general circulation model with a realistic quasi-biennial oscillation. Journal of Geophysical Research-Atmospheres, 109(D3), D03112. https://doi.org/10.1029/2003JD003699

16. Stenchikov, G., Hamilton, K., Stouffer, R. J., Robock, A., Ramaswamy, V., Santer, B., & Graf, H.-F. (2006). Arctic Oscillation response to volcanic eruptions in the IPCC AR4 climate models. Journal of Geophysical Research, 111(D7). https://doi.org/10.1029/2005JD006286

17. Stenchikov, G., Ukhov, A., Osipov, S., Ahmadov, R., Grell, G., Cady-Pereira, K., Mlawer, E., and Iacono, M. (2021). How does a Pinatubo-size Volcanic Cloud Reach the Middle Stratosphere? Journal of Geophysical Research: Atmospheres.

18. Toohey, M., Kru ̈ger, K., Bittner, M., Timmreck, C., and Schmidt, H. (2014). The impact of volcanic aerosol on the Northern Hemisphere stratospheric polar vortex: mechanisms and sensitivity to forcing structure. Atmospheric Chemistry and Physics, 14(23):13063–13079.